# FreeViS: Training-free Video Stylization with Inconsistent References

**Jiacong Xu**[1]    **Yiqun Mei**[2]    **Ke Zhang**[1]    **Vishal M. Patel**[1]
[1] Johns Hopkins University    [2] Adobe Research
{jxu155, kzhang99, vpatel36}@jhu.edu ymei@adobe.com
https://xujiacong.github.io/FreeViS/

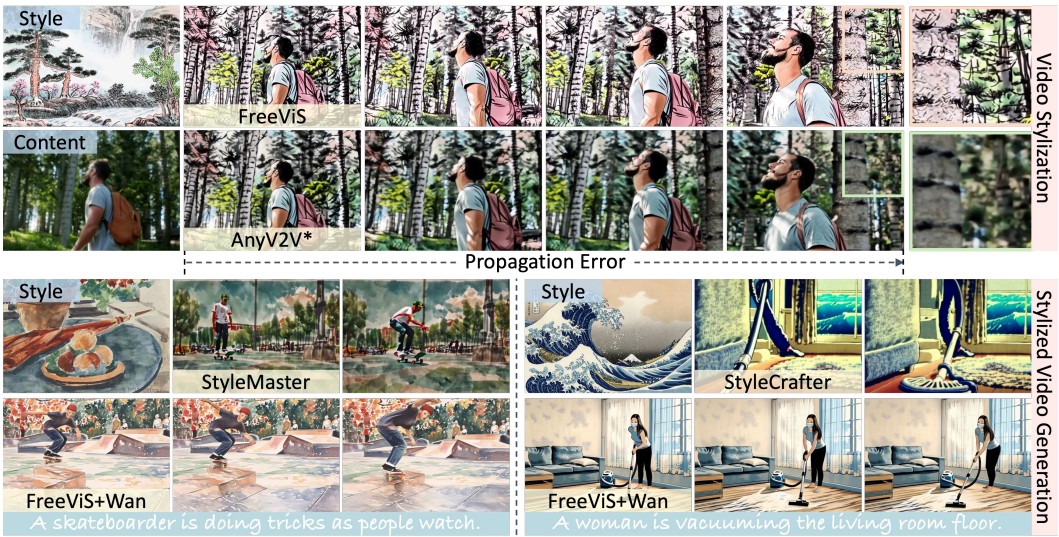

Figure 1: Previous works (e.g., AnyV2V Ku et al. (2024)) suffer from propagation errors inherent to their single-reference inputs. Combined with a text-to-video model Wan et al. (2025), FreeViS outperforms existing methods Ye et al. (2025); Liu et al. (2024a) on stylized video generation.

## ABSTRACT

Video stylization plays a key role in content creation, but it remains a challenging problem. Naïvely applying image stylization frame-by-frame hurts temporal consistency and reduces style richness. Alternatively, training a dedicated video stylization model typically requires paired video data and is computationally expensive. In this paper, we propose FreeViS, a training-free video stylization framework that generates stylized videos with rich style details and strong temporal coherence. Our method integrates multiple stylized references to a pretrained image-to-video (I2V) model, effectively mitigating the propagation errors observed in prior works, without introducing flickers and stutters. In addition, it leverages high-frequency compensation to constrain the content layout and motion, together with flow-based motion cues to preserve style textures in low-saliency regions. Through extensive evaluations, FreeViS delivers higher stylization fidelity and superior temporal consistency, outperforming recent baselines and achieving strong human preference. Our training-free pipeline offers a practical and economic solution for high-quality, temporally coherent video stylization.

## 1 INTRODUCTION

Style transfer has been a long-standing research topic in computer vision, with applications spanning art, education, advertising, and entertainment. In recent years, tremendous progress has been made in image style transfer Gatys et al. (2016); Huang & Belongie (2017); Liu et al. (2021); Chung et al. (2024); Xu et al. (2025b) , driven by advances in deep learning architectures and generative

modeling. While the focus has been on still images, videos have become a more popular medium for everyday communication in recent years. Yet video stylization remains significantly underexplored compared to its image-based counterpart. As content distribution changes across time, naïvely applying image stylization frame by frame leads to severe flickering artifacts, resulting in poor temporal consistency. Even though advanced temporal smoothing techniques Duan et al. (2023) can mitigate flickering artifacts, they often do so at the expense of losing style richness, resulting in overly smoothed textures and reduced visual plausibility.

Previous video stylization works Wang et al. (2020); Pande & Sharma (2023) have attempted to improve temporal consistency by modifying convolutional architectures. However, their quality remains inferior to that of recent diffusion-based image stylization approaches. More recently, several diffusion-based methods Liu et al. (2024a); Ye et al. (2025) have been proposed for stylized text-to-video (T2V) generation, but these approaches are not directly applicable to video-to-video (V2V) editing. In addition, our experiments also found that neither recent unified video editing frameworks Jiang et al. (2025) nor text-driven editing methods Geyer et al. (2024) are able to perfectly transfer style across videos. Beyond these works, directly fine-tuning a video diffusion model (*e.g.*, DiT-based architectures) is also impractical, as it demands large-scale original–stylized video pairs, which are difficult to obtain. Simply adopting the perceptual training strategy, as in image stylization Liu et al. (2021), cannot ensure temporal consistency across frames. Moreover, substantial computational resources are required for fully finetuning.

On the other hand, reference-based video editing methods, such as AnyV2V Ku et al. (2024), can leverage image stylization models to stylize the first frame, and then propagate it to the rest of the frames using pretrained image-to-video (I2V) models. In this way, no training or only limited training Ouyang et al. (2024) is required. However, due to the absence of stylized videos in the training phase of the base I2V model, it fails to properly handle the reference frame as it is out of the training distribution (see Appendix A.7), thus incapable of parsing and propagating style patterns from the first edited frame. As Figure 1 shows, these reference-based methods struggle to transfer style patterns to novel content in subsequent frames, resulting in a pronounced *propagation error*.

In this paper, we aim to tackle video stylization in a training-free manner, *i.e.* by inverting a diffusion model, to achieve comprehensive style transfer while preserving temporal coherence. An intuitive solution to address *propagation error* is to provide the base model with multiple reference frames across the entire video. However, naïvely concatenating additional reference frames with noise latent leads to severe flickering and stuttering artifacts in the results. To address this, we introduce isolated attention with a carefully designed masking strategy to mitigate stylization inconsistencies, together with a novel approach to inject temporal dynamics into the reference latents. We also found that the inverted noises alone are insufficient to accurately reconstruct original video content, often missing structures in stylized results. To this end, we propose a strategy to extract high-frequency components from the reconstruction compensation of PnP inversion Ju et al. (2024) to compensate and constrain spatial layouts and motion trajectories. Lastly, in plain regions with few salient features, style textures may vanish under large camera motion. To overcome this, we propose a method that leverages optical flow to constrain diffused attention areas in associated frames.

The contributions of this paper are summarized as follows: (1) We propose a novel reference-based and training-free framework: FreeViS, for video stylization, which effectively addresses the propagation error inherent in existing methods; (2) We conduct extensive experiments demonstrating the effectiveness of our design and showing that FreeViS achieves higher stylization quality compared to prior approaches; (3) We show that, when combined with existing T2V models, FreeViS also achieves stylized T2V generation with performance comparable to recent training-based methods.

## 2 OBSERVATIONS

### 2.1 FREQUENCY ANALYSIS OF INTERMEDIATE LATENTS

Since the inverted noises alone cannot accurately reconstruct the videos (Appendix A.7), we adopt PnP Inversion Ju et al. (2024) to progressively align the denoising trajectory with the inversion path. Building on this framework, we investigate the influence of low-frequency (LF) and high-frequency (HF) components of intermediate latents along the denoising process (Appendix A.8). Recent image editing studies Yang et al. (2023); Xu et al. (2025b) have shown that the HF components of the initial noise primarily determine the spatial layout of the generated images. Here, we observe a similar

phenomenon in I2V models: *LF latents primarily govern the appearance and color distribution of the generated video, whereas HF components encode layout and motion cues.*

## 2.2 CROSS-FRAME ATTENTION

Recent advances in video generation Kong et al. (2024); Wan et al. (2025) have seen a shift in architectural design from UNet-based models Blattmann et al. (2023) to Transformer-based DiT frameworks Peebles & Xie (2023). For finer-grained analysis of attention patterns, we decompose the cross-frame attention in the I2V Wan model into temporal and spatial components . To visualize temporal attention, we spatially average the attention values across each frame. For spatial attention, we select a point on the query frame and record the distribution of attention intensities across other frames. As shown in Figure 2, the model exhibits an auto-causal attention pattern, with lower attention values in the upper triangular region of the temporal attention matrices. Notably, the second frame (immediately following the reference frame) consistently receives high attention throughout the entire denoising process. This suggests that the reference frame can influence the generation of all subsequent frames indirectly, by strongly guiding the denoising of the second frame.

During late stages of denoising, the model concentrates on refining local details, such as textures, resulting in more accurate pixel-level correspondences compared to earlier stages that primarily require global information for layout establishment. This is reflected in the sharper and more focused high-attention regions. However, when attending to frames that are temporally distant from the query frame, the pixel-level attention tends to diffuse into incorrect areas, likely due to increased motion (camera or objects) between distant frames relative to adjacent ones. Note that accurate pixel correspondences are crucial for preserving the consistency of local stylized textures or strokes in reference-based video stylization. Therefore, an external control strategy is required to regularize the attention area for better appearance preservation.

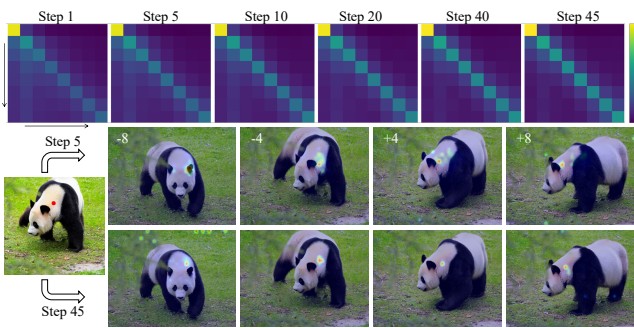

Figure 2: Visualization of cross-frame temporal (upper) and spatial (lower) attentions in different timesteps.

## 3 METHOD

The overall architecture of the proposed FreeViS pipeline is illustrated in Figure 3. Our framework builds upon a pretrained I2V diffusion model, which serves as the backbone. Given an arbitrary style image, stylized references are generated via an image style transfer model applied to several selected content video frames. We leverage inversion to recover the denoising trajectory and initial noise. The pipeline comprises two branches: reconstruction and stylization. Each branch is conditioned on the corresponding selected references. The reconstruction branch provides query, key, and value matrices to the stylization branch in every DiT block. Parameters specific to the reconstruction and stylization branches are denoted by superscripts $r$ and $s$, respectively. The subscript $R$ denotes parameters associated with additional references beyond the first frame. After each denoising step, the high-frequency components of the difference between the target latent $\mathbf{x}_t$ and reconstruction latent $\mathbf{x}_t^r$ are added to the stylization latent $\mathbf{x}_t^s$. The value matrices $\mathbf{v}_R^s$ projected from additional stylized references are injected with dynamic clues, as formalized in Equation 4.

### 3.1 INDIRECT HIGH-FREQUENCY COMPENSATION

We adopt PnP Inversion Ju et al. (2024) to provide additional guidance throughout the denoising process. Specifically, the full denoising trajectory $[\mathbf{x}_t, \mathbf{x}_{t-1}, \ldots, \mathbf{x}_0]$ is cached during inversion and treated as the target latents. In the original PnP framework, the difference between the target latent $\mathbf{x}_t \in \mathbb{R}^{s \times c \times h \times w}$, where $s$ is the number of encoded latent maps, and the current reconstruction latent $\mathbf{x}_t^r \in \mathbb{R}^{s \times c \times h \times w}$ is computed and applied to both the reconstruction and editing branches, enabling near-exact reconstruction. However, we observe that this strong correction directly drives

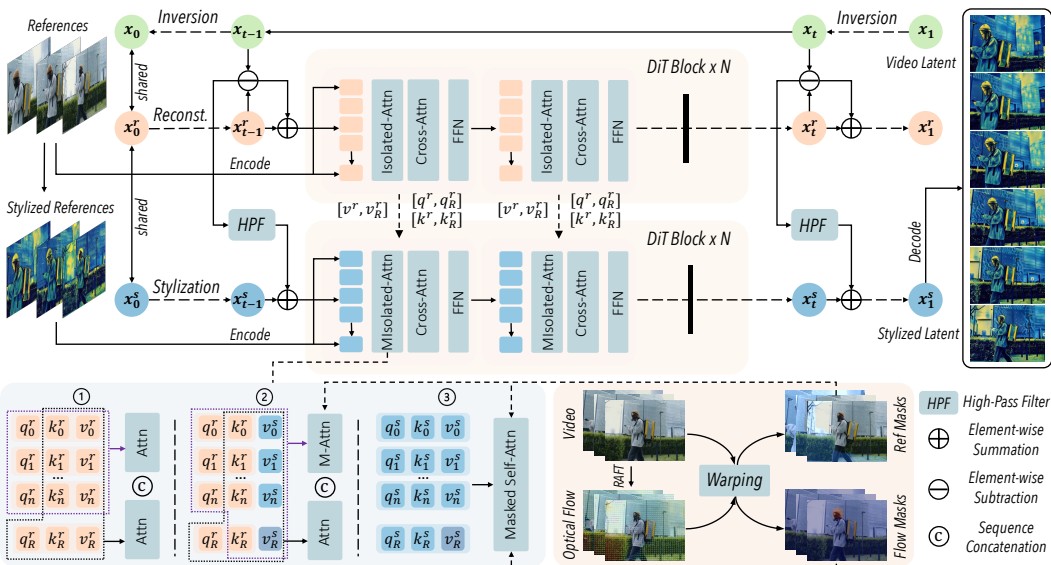

Figure 3: Overview of the FreeViS pipeline. **Isolated-Attn** indicates the ① mode, while **MIsolated-Attn** includes ② and ③ attention modes. Optical flow, extracted using RAFT Teed & Deng (2020), generates reference and flow masks for masked attention in attention modes ② and ③.

all the information of the stylized latent toward the original target, resulting in undesirable color distribution shifts from the stylized video back toward the original content.

As analyzed in Section 2.1, low-frequency (LF) latents primarily control appearance and color, while high-frequency (HF) components encode layout and motion. This motivates our Indirect High-frequency Compensation (IHC) strategy, which injects only HF differences into the stylized latents, preserving the stylized appearance while correcting structural inconsistencies. In the reconstruction branch, full compensation is applied to recover video content: $\mathbf{x}_t^r = \lambda \cdot (\mathbf{x}_t - \mathbf{x}_t^r) + \mathbf{x}_t^r$, where $\lambda$ is a hyperparameter linearly decaying over timesteps. To further align the color distribution, we firstly apply AdaIN Huang & Belongie (2017) on both $\mathbf{x}_t$ and $\mathbf{x}_t^r$ before calculating their difference:

$$\mathcal{T}(\mathbf{x}_t) = \sigma(\mathbf{x}_t^s) \left( \frac{\mathbf{x}_t - \mu(\mathbf{x}_t)}{\sigma(\mathbf{x}_t)} \right) + \mu(\mathbf{x}_t^s), \qquad \mathcal{T}(\mathbf{x}_t^r) = \sigma(\mathbf{x}_t^s) \left( \frac{\mathbf{x}_t^r - \mu(\mathbf{x}_t^r)}{\sigma(\mathbf{x}_t^r)} \right) + \mu(\mathbf{x}_t^s), \quad (1)$$

where $\mathbf{x}_t^s$ refers to the stylization latents in current timestep $t$, and $\mu(\cdot)$ and $\sigma(\cdot)$ denote channel-wise mean and standard deviation, respectively. We then compute the Fast Fourier Transform (FFT) of the reconstruction difference along spatial dimensions to obtain the frequency representation. A low-pass filter $H_{LP}$ is applied to isolate HF components, which are subsequently transformed back via inverse FFT (iFFT) and added to the stylized latent:

$$\mathbf{x}_t^s = \lambda \cdot \mathcal{F}^{-1} \left( \mathcal{F} \left( \mathcal{T}(\mathbf{x}_t) - \mathcal{T}(\mathbf{x}_t^r) \right) \cdot (1 - H_{LP}) \right) + \mathbf{x}_t^s, \qquad (2)$$

where $\mathcal{F}(\cdot)$ and $\mathcal{F}^{-1}(\cdot)$ denote FFT and iFFT. Empirically, the cutoff frequency of the low-pass filter is set to be 0.2 for the best trade-off between style fidelity and content reconstruction. The experimental results show that IHC successfully preserves the stylized color and texture while enhancing spatial consistency and motion fidelity, particularly in scenes with significant camera movement or when later frames differ substantially from the first frame.

## 3.2 Additional Inconsistent References

Previous I2V-based video editing methods Ku et al. (2024); Ouyang et al. (2024) typically rely on a single stylized first frame to guide the stylization of the entire video. While effective when content changes are minimal, this approach struggles to propagate style features to regions that differ significantly from the first frame. An intuitive solution is to incorporate multiple stylized references. However, existing advanced I2V models support only a single reference input. Naively concatenating additional reference tokens with reconstruction or stylization noise tokens prior to self-attention often results in significant **flickering** and **stuttering** in the generated videos.

To enable multi-reference inputs in the reconstruction branch, we propose the Isolated-Attn strategy, which isolates the influence of auxiliary references $\mathbf{x}_R^r$, thereby preventing interference with

the original denoising schedule. Specifically, the reconstruction tokens $\mathbf{x}^r$ undergo standard self-attention, while the reference tokens $\mathbf{x}_R^r$ attend to both reconstruction and reference keys and values. This design allows reference tokens to evolve in sync with the denoising process, mimicking full self-attention behavior. Denote the Query, Key, and Value matrices projected from $\mathbf{x}^r$ and $\mathbf{x}_R^r$ as $\{\mathbf{Q}^r, \mathbf{K}^r, \mathbf{V}^r\}$ and $\{\mathbf{Q}_R^r, \mathbf{K}_R^r, \mathbf{V}_R^r\}$, respectively. Then, this mechanism can be defined as:

$$\mathbf{Out}^r = \mathcal{A}(\mathbf{Q}^r, \mathbf{K}^r, \mathbf{V}^r) \oplus \mathcal{A}(\mathbf{Q}_R^r, \mathbf{K}^r \oplus \mathbf{K}_R^r, \mathbf{V}^r \oplus \mathbf{V}_R^r), \quad (3)$$

where $\mathcal{A}(\cdot)$ refers to attention operation, and $\oplus$ denotes sequence-wise token concatenation.

Differently, in the stylization branch, full information exchange among all tokens is essential for comprehensive style transfer. However, since the additional stylized references $\mathbf{x}_R^s$ are independently encoded, the value matrices $\mathbf{V}_R^s$ projected from $\mathbf{x}_R^s$ lack dynamic information, which causes the **stuttering** issue in adjacent frames. We observe that appearance information can be isolated and exchanged between videos with shared dynamics (see details in Appendix A.5). Since the dynamic information is shared between stylization values $\mathbf{V}^s$ and reconstruction values $\mathbf{V}^r$, we decouple the dynamic residual by computing their differences relative to their respective references (only contain appearance), and inject only the dynamic component into $\mathbf{V}_R^s$. This is implemented as:

$$\mathbf{V}_R^s = \mathbf{V}_R^s + \xi \cdot (\mathbf{V}^s[i_R] - \mathbf{V}_R^s) + (1 - \xi) \cdot (\mathbf{V}^r[i_R] - \mathbf{V}_R^r), \quad (4)$$

where $i_R$ denotes the positional indices of the references, and $\xi$ increases linearly from 0 to 1 over the timesteps. Early in the denoising process, the model relies more on dynamics from the reconstruction branch; toward the end, $\mathbf{V}_R^s$ converges to $\mathbf{V}^s[i_R]$ to ensure consistent final outputs.

Since inconsistencies naturally exist across stylized references, the same region may exhibit varying appearances, resulting in time-variant stylization artifacts, such as **flickering**. To address this issue, we leverage optical flow extracted from the content video to identify regions already covered by earlier references and mask these regions in attention, thereby resolving appearance conflicts. We employ RAFT Teed & Deng (2020) to compute optical flow and trace pixel correspondences from the first reference frame to subsequent reference frames. The reference mask $M_{Ref}$ is constructed such that a position is marked as `False` if the corresponding pixel is reachable from previous references(see Appendix A.3). Masked attention $\mathcal{A}_{Masked}(\cdot)$ is implemented to incorporate $M_{Ref}$:

$$\mathbf{Out}_1^s = \mathcal{A}_{Masked}(\mathbf{Q}^s, \mathbf{K}^s \oplus \mathbf{K}_R^s, \mathbf{V}^s \oplus \mathbf{V}_R^s, M_{Ref}) \oplus \mathcal{A}(\mathbf{Q}_R^s, \mathbf{K}^s \oplus \mathbf{K}_R^s, \mathbf{V}^s \oplus \mathbf{V}_R^s), \quad (5)$$

where $\{\mathbf{Q}^s, \mathbf{K}^s\}$ and $\{\mathbf{Q}_R^s, \mathbf{K}_R^s\}$ refer to the Query and Key matrices projected by $\mathbf{x}^s$ and $\mathbf{x}_R^s$. Inspired by prior UNet-based approaches that forward convolution and attention features from reconstruction to editing branches, we leverage QK-Sharing of the reconstruction queries and keys to enable more flexible attention operations in the stylization branch. These queries and keys define spatial-temporal correspondences across the content video, which are critical for effective style propagation and temporal consistency. Thus, we replace the query and value in Equation 5:

$$\mathbf{Out}_2^s = \mathcal{A}_{Masked}(\mathbf{Q}^r, \mathbf{K}^r \oplus \mathbf{K}_R^r, \mathbf{V}^s \oplus \mathbf{V}_R^s, M_{Ref}) \oplus \mathcal{A}(\mathbf{Q}_R^r, \mathbf{K}^r \oplus \mathbf{K}_R^r, \mathbf{V}^s \oplus \mathbf{V}_R^s) \quad (6)$$

Due to the auto-causal nature of the base model, the style of the generated video is primarily determined by the first reference frame, while subsequent references act as supplementary sources.

### 3.3 Explicit Optical Flow Guidance

When the camera or object motion is significant, we observe disappearing or time-varying style textures in plain areas. We attribute this issue to inaccurate attention maps between temporally distant frames, especially in regions with little salient visual features, as discussed in Section 2.2. To address this, we introduce Explicit Optical-flow Guidance (EOG), which provides near pixel-level correspondence via optical flow. We first compute forward and backward optical flows and trace every pixel across frames. If a pixel $p_{i,j}^s$ in frame $s$ maps to location $p_{m,n}^t$ in frame $t$, the flow mask $M_{Flow} \in \{0,1\}^{T \times h \times w \times T \times h \times w}$ is set to `True` at index $(s, i, j, t, m, n)$ (see Appendix A.4). We further apply dilation to compensate for estimation errors of the optical flow.

Using this mask, we perform masked attention on stylization and corresponding reference tokens:

$$\mathbf{Out}_3^s = \mathcal{A}_{Masked}(\mathbf{Q}^s \oplus \mathbf{Q}_R^s, \mathbf{K}^s \oplus \mathbf{K}_R^s, \mathbf{V}^s \oplus \mathbf{V}_R^s, M_{Flow} \wedge M_{Ref}), \quad (7)$$

where $\wedge$ denotes a logical AND, ensuring that only consistent regions across flow and reference masks contribute. This design constrains the attention area of the source pixel on every frame,

| Method | Stylization Quality | | | | Video Consistency | | | HP ↑ |
|---|---|---|---|---|---|---|---|---|
| | CSD Score ↑ | ArtFID ↓ | FID ↓ | LPIPS ↓ | SC ↑ | MS ↑ | FC ↓ | |
| Reference | 0.508 | 31.62 | 20.28 | 0.486 | 0.918 | 0.986 | 0.000 | - |
| TokenFlow | 0.111 | 37.87 | 27.94 | 0.309 | **0.915** | 0.976 | 1.092 | 2.179 |
| VACE | 0.138 | 35.53 | 27.77 | 0.240 | 0.910 | **0.984** | **0.554** | 2.895 |
| I2VEdit | 0.331 | 38.72 | 22.53 | 0.653 | 0.738 | 0.975 | 2.074 | 2.538 |
| AnyV2V | 0.267 | 35.84 | 23.52 | 0.471 | 0.753 | 0.961 | 1.715 | 2.443 |
| AnyV2V* | 0.270 | 34.81 | 27.59 | **0.218** | 0.675 | 0.983 | 1.103 | 3.372 |
| Ours | **0.448** | **33.46** | **21.62** | 0.479 | 0.898 | 0.978 | 0.641 | **4.113** |

Table 1: Quantitative results for video stylization. "SC", "MS", "FC", and "HP" denote Style Consistency, Motion Smoothness, Flow Consistency, and Human Preference, respectively. We use the stylized references and original video to anchor the quality of stylization and video consistency.

explicitly controlling the appearance to follow the video dynamics. Empirically, we find that using queries and keys from stylization tokens outperforms those from the reconstruction branch. Finally, the outputs from all three attention modes are aggregated before cross-attention as:

$$\mathbf{Out}^s = (1 - \beta - \gamma) \cdot \mathbf{Out}_1^s + \beta \cdot \mathbf{Out}_2^s + \gamma \cdot \mathbf{Out}_3^s, \tag{8}$$

where $\beta$ and $\gamma$ are two hyperparameters, controlling the impact of the inconsistent references and the strength of the explicit appearance guidance. $\gamma$ is set to a non-zero value only in the final stage of the denoising process, when the model primarily focuses on local texture refinement.

**Cross-Attention.** The base model further encodes the reference image using CLIP Radford et al. (2021) to provide high-level semantic guidance via cross-attention. To incorporate multiple references, we concatenate the CLIP features from all reference frames and apply QK-Sharing in the cross-attention layer, enhancing the injection of language-aligned information. As CLIP features contain limited spatial structure, reference and flow masks are not applied.

**Reference Arrangement.** In principle, all the frames can be stylized and used as references in FreeViS for comprehensive style coverage. Nevertheless, increasing the number of references significantly raises computational and memory costs. Considering that most advanced I2V models are limited to short videos (typically around 81 frames), we select the first, middle, and last frames as references, an empirically effective choice for achieving high-quality stylization in most cases. Each reference token is assigned the same positional embedding as its corresponding frame latent to guarantee correct spatial and temporal propagation of stylized features.

## 4 EXPERIMENTS

This section is divided into two major parts: Video Style Transfer and Stylized T2V Generation. All the stylized references for FreeViS and other reference-based video editing models are provided by InstantStyle-plus Wang et al. (2024b). We deliver the results of FreeViS combined with various image stylization methods and different video diffusion base models in Appendix A.10 and A.18.

### 4.1 VIDEO STYLE TRANSFER

**Dataset & Metrics.** We curated 200 online videos covering human activities, natural and urban scenes, animals, and transportation. Style images were sourced from WikiArt Tan et al. (2018) and web collections. For evaluation, we use the CSD Score Somepalli et al. (2024) to assess frame–style similarity, and ArtFID Wright & Ommer (2022) to jointly evaluate stylization quality and content preservation via FID and LPIPS. Motion smoothness is measured with VBench Huang et al. (2024), and optical flow consistency between the original and stylized videos with End-Point Error (EPE) Barron et al. (1994). Finally, we introduce Style Consistency, which quantifies the similarity between the first frame and subsequent frames based on style features extracted by Somepalli et al. (2024). We conduct a human preference evaluation to assess both the visual plausibility and the stylization quality of the generated videos. Each examiner is asked to assign a score from 1 to 5 for both aspects, and the final rating is obtained by averaging the two scores.

**Baselines.** We compare against various baselines: reference-based video editing (AnyV2V Ku et al. (2024), I2VEdit Ouyang et al. (2024)), text-driven editing (TokenFlow Geyer et al. (2024)), unified

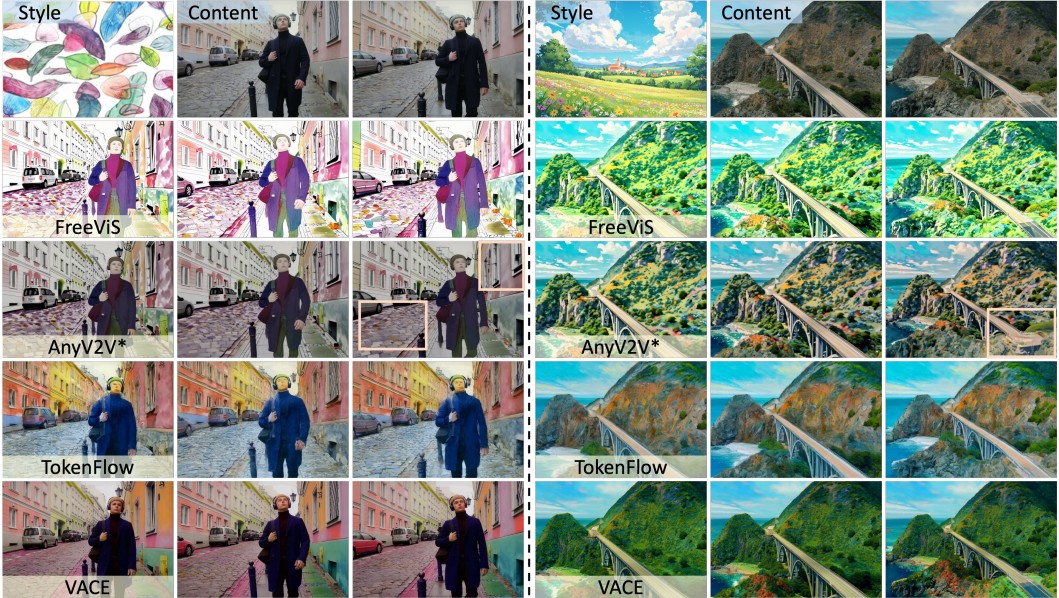

Figure 4: Qualitative comparison of FreeViS with other video editing methods on video stylization. The areas inside the bounding boxes show missing style textures and incorrect reconstruction.

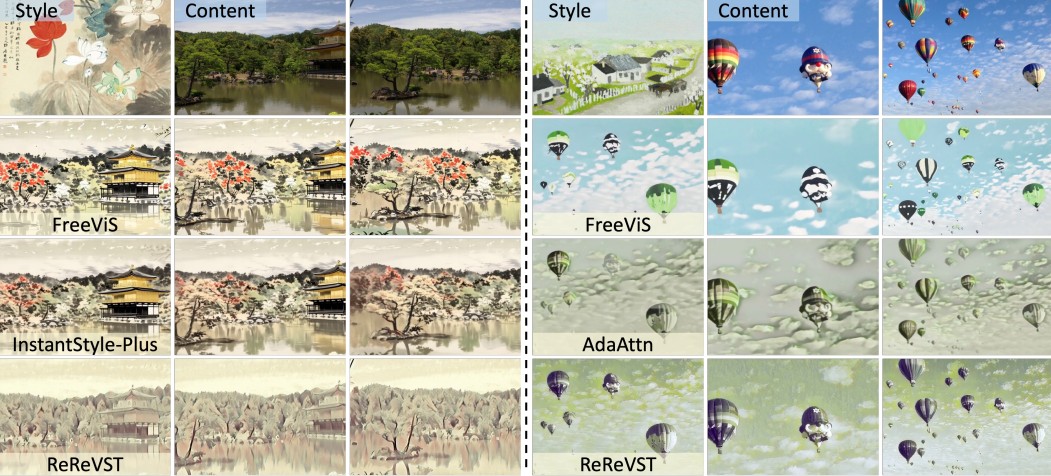

Figure 5: Qualitative comparison of FreeViS with previous video and image stylization methods. The flickering issue of the image stylization methods can be observed in the supplemented video.

video editing (VACE Jiang et al. (2025)), classic video style transfer (ReReVST Wang et al. (2020), ViSt3D Pande & Sharma (2023)), and image style transfer with temporal smoothing Duan et al. (2023) (AdaAttn Liu et al. (2021), InstantStyle-plus Wang et al. (2024b)). Since the code for V2V implementation of StyleMaster Ye et al. (2025) has not been released, we omit its results currently. We re-implement AnyV2V, named by AnyV2V*, using the same base model as FreeViS for a fair comparison. Note that among all the models, only AnyV2V and TokenFlow are training-free.

**Results.** Qualitative comparisons are presented in Figures 4 and 5. VACE demonstrates strong video consistency (Table 1) but is limited to color modification, struggling to capture and transfer complex style textures. Similarly, TokenFlow faces challenges in style representation, primarily due to the limitations of text-based descriptions, which parallel the issues observed with VACE. Since the modifications from VACE and TokenFlow on the original videos mainly focus on color changes instead of new texture generation, the video consistency scores are naturally better than other methods and close to the reference anchor. Also, the MS and FC are computed using flow-estimation models pre-trained on natural videos. However, the stylized videos can be considered out-of-distribution samples, leading to inaccurate flow estimates and degraded scores. The original AnyV2V and I2VEdit models fail to reconstruct the video when there are substantial content

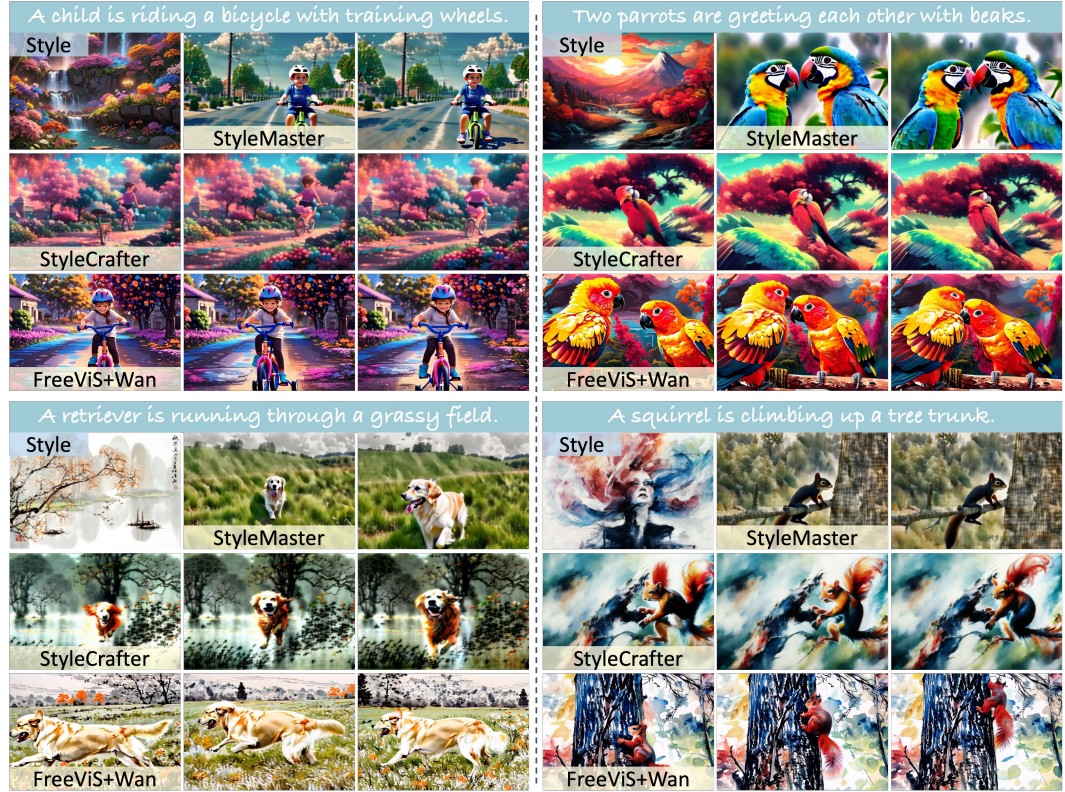

Figure 6: Qualitative comparison of FreeViS with previous methods for stylized video generation. The difference can be easily observed in the supplemented video.

| Method | Stylization Quality | | Video Generation Quality | | | | | HP ↑ |
|---|---|---|---|---|---|---|---|---|
| | CSD Score ↑ | FID ↓ | CLIP-Text ↑ | DQ ↑ | MS ↑ | BC ↑ | IQ ↑ | |
| StyleCrafter | **0.515** | **22.62** | 0.211 | 0.368 | 0.965 | **0.951** | 0.578 | 2.83 |
| StyleMaster | 0.221 | 26.04 | 0.243 | 0.123 | **0.985** | 0.945 | 0.667 | 2.55 |
| Ours+Wan | 0.437 | 24.63 | **0.264** | **0.509** | 0.980 | 0.941 | **0.691** | **3.97** |

Table 2: Quantitative results for stylized video generation. "DQ", "MS", "BC", "IQ", and "HP" denote Dynamic Quality, Motion Smoothness, Background (CLIP) Consistency, Imaging Quality, and Human Preference, respectively. A smaller DQ can naturally lead to better MS and BC.

changes between the first and last frames (see Appendix A.17), likely due to the weak priors of their pre-trained base models. While AnyV2V* significantly mitigates this issue, it still suffers from propagation errors: early-frame styles are not consistently transferred to later frames, resulting in the lowest style consistency score. In contrast, FreeViS effectively preserves stylistic features across the entire sequence, achieving the highest stylization quality and near-optimal temporal consistency.

Classical video stylization methods, which typically rely on non-diffusion architectures, offer strong temporal consistency but suffer from limited stylization quality. In contrast, image-based stylization methods, while capable of richer stylization, exhibit significant frame-wise inconsistencies. This leads to over-flattened style textures and persistent flickering artifacts, even after temporal smoothing. FreeViS overcomes these limitations by leveraging a pretrained video diffusion framework that effectively balances high-quality stylization with robust temporal coherence.

## 4.2 STYLIZED T2V GENERATION

**Dataset & Metrics & Baselines.** We utilize Qwen3 Yang et al. (2025a) to generate 200 text prompts describing the content of target videos, covering a broad range of commonly encountered real-world scenarios. The style images used are identical to those described in the previous section. As no ground-truth videos are available, we adopt four evaluation metrics from VBench Huang et al. (2024): Dynamic Quality, Motion Smoothness, Background Consistency, and Imaging Quality, to

Figure 7: Ablation study of FreeViS. The upper images are generated from the full model.

assess the visual quality of the generated videos. Additionally, CLIP-Text similarity is employed to evaluate video-prompt alignment. For assessing style consistency and fidelity, we report the CSD Score and FID, which measure the similarity between video frames and style images. Human preference is also employed to assess the prompt alignment, stylization quality, and visual plausibility. We compare our approach against two state-of-the-art baselines Liu et al. (2024a); Ye et al. (2025): StyleCrafter and StyleMaster. For stylized T2V generation, we first generate base videos using Wan2.1 Wan et al. (2025), followed by applying FreeViS for video stylization.

**Results.** The qualitative and quantitative results for stylized video generation are presented in Figure 6 and Table 2. StyleCrafter effectively transfers style into the generated videos; however, its outputs often lack dynamic realism and occasionally fail to align with the input prompts. Since the camera and object motions of the generated videos from StyleCrafter are relatively small (see the comparison results in the supplemented video), the calculated consistency scores are higher than those of other methods. In contrast, StyleMaster better adheres to the textual prompts but exhibits limited stylization capability. The combination of FreeViS and the Wan model achieves the best trade-off between stylization fidelity and content alignment, producing videos that are both visually compelling and semantically accurate, as reflected by the highest human preference scores. Note that the stylization performance of FreeViS is inherently upper-bounded by the underlying image style transfer method it employs.

### 4.3 ABLATION STUDY

FreeViS primarily comprises three key components: IHC for layout reconstruction, Extra References for enforcing style consistency, and EOG for texture preservation in plain areas. The effectiveness of each component is illustrated in Figure 7. Without IHC, the model fails to accurately reconstruct detailed scene layouts, leading to structural artifacts, as evidenced by incorrect roof reconstruction. Incorporating additional reference frames helps maintain consistent style features throughout the video, significantly enhancing visual coherence. In the example, the last frame shows only a color shift and struggles to convey stylistic textural details. Furthermore, the EOG module mitigates the loss of style textures in visually homogeneous regions of the original video, thereby improving flow consistency and overall stylization quality.

## 5 RELATED WORK

**Video Diffusion.** Diffusion-based video generation is advancing rapidly, with significant progress in quality Agarwal et al. (2025); Yang et al. (2025c); Blattmann et al. (2023), efficiency Xi et al. (2025); Zhang et al. (2025a); Kahatapitiya et al. (2024), controllability Zhang et al. (2025b); Bai et al. (2025a); Ren et al. (2025), and extrapolation Zhao et al. (2025); Dalal et al. (2025); Mao et al. (2024). Unlike early UNet-based architectures Blattmann et al. (2023); Guo et al. (2024); Mei & Patel (2023), which employ separate layers to model spatial and temporal dependencies, recent advances Kong et al. (2024); Yang et al. (2025c); Wan et al. (2025) adopt the full self-attention layers of DiT Peebles & Xie (2023) to jointly capture pixel-level coherence.

**Video Editing.** Leveraging the strong priors of video diffusion models, recent editing methods have significantly outperformed earlier approaches based on optical flow Xu et al. (2019), GANs Tzaban et al. (2022); Frühstück et al. (2023), and image diffusion Ceylan et al. (2023); Geyer et al. (2024); Feng et al. (2024). Unlike image editing, which mainly enforces spatial coherence Brooks et al. (2023a;b), video editing additionally requires preserving temporal consistency Yang et al. (2025b). Prior work has addressed diverse tasks such as inpainting Bian et al. (2025); Guo et al. (2025), content swapping Liu et al. (2024b); Tu et al. (2025); Zhang et al. (2025c), and interactive control

Deng et al. (2024); Ma et al. (2025). VACE Jiang et al. (2025) proposes a unified framework for video generation and editing with a mask-guided condition unit, while AnyV2V Ku et al. (2024) and I2VEdit Ouyang et al. (2024) leverage the first edited frame as guidance for pre-trained I2V models.

**Style Transfer.** Neural style transfer Gatys et al. (2016) has progressed with deep architectures Huang & Belongie (2017); Liu et al. (2021); Deng et al. (2022) and the generative models Azadi et al. (2018); Zhang et al. (2023); Wang et al. (2023). Diffusion models further improve semantic alignment between style and content Wang et al. (2024b); Xing et al. (2024), inspiring training-free methods Wang et al. (2024a); Rout et al. (2025); Chung et al. (2024); Xu et al. (2025b) that surpass traditional techniques. By contrast, video stylization remains underexplored. ViSt3D Pande & Sharma (2023) employs 3D CNNs for temporal consistency, StyleMaster Ye et al. (2025) finetunes a video diffusion model with CLIP-guided cross-attention, and StyleCrafter Liu et al. (2024a) enables style-conditioned text-to-video generation. Nonetheless, these approaches still lag behind image style transfer in quality, with temporal consistency being a key challenge.

## 6 CONCLUSION

In this paper, we introduce a training-free video stylization framework, FreeViS, which successfully addresses the propagation errors inherent in prior works. FreeViS leverages IHC to constrain the layout and motion. To enable additional reference inputs without requiring extra fine-tuning, we propose isolated attention with masking and dynamics injection. Furthermore, we propose the EOG strategy to preserve style textures in plain areas. Extensive experiments are conducted to validate the effectiveness of our design and the superior performance of FreeViS over other methods.

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

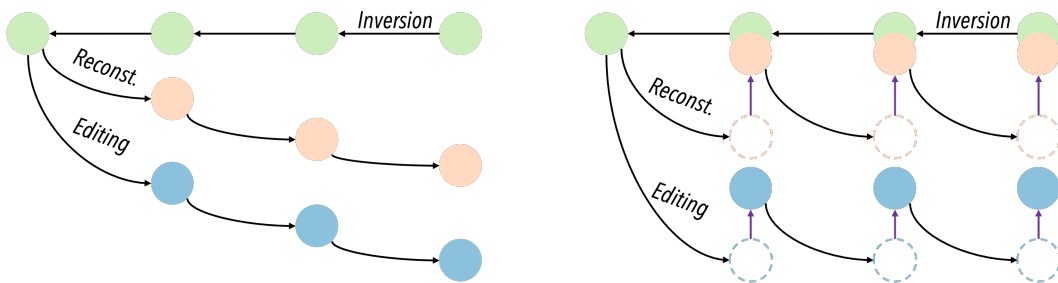

Figure 8: Latent movement along the inversion and denoising process for reconstruction and editing. Left: Conventional Inversion; Right: PnP Inversion. Purple arrows refer to the compensation.

## A    APPENDIX

### A.1    PRELIMINARY: PNP INVERSION

Different from conventional inversion-based editing methods, PnP inversion Ju et al. (2024) records the full denoising trajectory $[\mathbf{x}_t, \mathbf{x}_{t-1}, \ldots, \mathbf{x}_0]$ (the *target latents*) during inversion, and explicitly injects the discrepancy between the reconstruction latent $\mathbf{x}_t^r$ and the target latent $\mathbf{x}_t$ into both the reconstruction and editing latents $\mathbf{x}_t^e$. This compensation mitigates the accumulated error that is otherwise amplified by classifier-free guidance, thereby steering both reconstruction and editing back toward the original denoising trajectory (Figure 8). The update rule can be expressed as:

$$\mathbf{x}_t^r \leftarrow \mathbf{x}_t^r + (\mathbf{x}_t - \mathbf{x}_t^r), \quad \mathbf{x}_t^e \leftarrow \mathbf{x}_t^e + (\mathbf{x}_t - \mathbf{x}_t^r) \tag{9}$$

In this paper, we observe that the inverted noise alone is insufficient to guarantee the correct reconstruction of the layout and motion in modern video diffusion models, and PnP inversion (combined with our IHC strategy) can successfully resolve this issue.

### A.2    IMPLEMENTATION DETAILS

Control signals such as depth, Canny edges, and tile images offer additional content guidance for generative models and have been widely adopted in both training-free and training-based style transfer approaches Xu et al. (2025b); Xing et al. (2024); Wang et al. (2024a); Ye et al. (2025). To enhance motion preservation, we adopt Wan2.1-Fun-Control (0.5B) as our pretrained base model, an adapted variant of Wan2.1 Wan et al. (2025) that can incorporate depth as a control signal. This base model positions the reference tokens at the beginning of the visual token sequence and supports only a single reference image, which serves as the initial frame of the generated video. One can freely change the base model to other video diffusion models, given the FreeViS framework. We utilize Euler Inversion with fixed-point iteration Xu et al. (2025a) to perform inversion under the Rectified Flow framework Lipman et al. (2023). The number of iterations can be adjusted based on the users' computational budgets. During inversion, we omit the reference latent and provide only the video depth map parsed by Video Depth Anything Chen et al. (2025) and a concise text prompt extracted using Qwen2.5-VL Bai et al. (2025b), which describes the content and dynamics of the input video.

### A.3    COMPUTATION OF REFERENCE MASKS

Let video frames be indexed by $t = 0, \ldots, T-1$ with pixel domain $\Omega_t \subset \mathbb{Z}^2$. We adopt RAFT Teed & Deng (2020) $\mathbf{f}_t : \mathbb{R}^2 \to \mathbb{R}^2$ to extract the forward optical flow from frame $t$ to $t+1$, and let $\Pi : \mathbb{R}^2 \to \Omega_t$ denote nearest-neighbor discretization to pixels.

$$\mathbf{p}_s = \mathbf{p}, \quad \mathbf{p}_{k+1} = \mathbf{p}_k + \mathbf{f}_k(\mathbf{p}_k) \quad (k = s, \ldots, t-1), \qquad \Phi_{s \to t}(\mathbf{p}) = \mathbf{p}_t \tag{10}$$

where $\Phi_{s \to t}$ denotes the forward propagation operator. Given a target frame $t$ and a source set $S \subseteq \{0, \ldots, T-1\}$, the set of pixels in frame $t$ covered by $S$ is computed by:

$$\mathcal{C}_t(S) = \left\{ \Pi\big(\Phi_{a \to t}(\mathbf{u})\big) \ : \ \mathbf{u} \in \Omega_a, \ a \in S \right\} \subseteq \Omega_t \tag{11}$$

With two given frame indices $i$ and $j$ ($0 < i < j \leq T - 1$), the two novel-region masks are:

$$M_i(\mathbf{p}) = \mathbb{1}\big[\mathbf{p} \notin \mathcal{C}_i(\{0\})\big], \qquad M_j(\mathbf{p}) = \mathbb{1}\big[\mathbf{p} \notin \mathcal{C}_j(\{0, i\})\big], \quad \mathbf{p} \in \Omega_t \tag{12}$$

This approach can be extended to more references with a pre-defined order.

A.4 Computation of Flow Masks

We construct a dense binary correspondence mask that records, for every source pixel in frame $s$, the (discretized) terminal location it reaches in any target frame $t$ when transported by optical flow. Let frames be indexed by $t = 0, \ldots, T-1$ with pixel domains $\Omega_t \subset \mathbb{Z}^2$. As in §A.3, RAFT Teed & Deng (2020) provides forward flows $\mathbf{f}_k : \mathbb{R}^2 \to \mathbb{R}^2$ from $k$ to $k+1$ (and analogously backward flows $\tilde{\mathbf{f}}_k$ from $k+1$ to $k$), and $\Pi_t : \mathbb{R}^2 \to \Omega_t$ denotes nearest-neighbor discretization onto frame $t$.

**Intermediate sampling and terminal validity.** Starting at $\mathbf{p}_s = \mathbf{u} \in \mathbb{R}^2$, we propagate *forward* from $s$ to $t \geq s$ by sampling the flow at the nearest valid pixel and updating the *continuous* state:

$$\hat{\mathbf{p}}_k = \Pi_k(\mathbf{p}_k), \qquad \mathbf{p}_{k+1} = \mathbf{p}_k + \mathbf{f}_k(\hat{\mathbf{p}}_k), \quad k = s, \ldots, t-1, \qquad \Phi_{s \to t}(\mathbf{u}) = \mathbf{p}_t \quad (13)$$

The *backward* transport for $t < s$ is defined analogously using $\tilde{\mathbf{f}}_k$:

$$\hat{\mathbf{p}}_{k+1} = \Pi_{k+1}(\mathbf{p}_{k+1}), \qquad \mathbf{p}_k = \mathbf{p}_{k+1} + \tilde{\mathbf{f}}_k(\hat{\mathbf{p}}_{k+1}), \quad k = s-1, \ldots, t, \qquad \tilde{\Phi}_{s \to t}(\mathbf{u}) = \mathbf{p}_t \quad (14)$$

Note that intermediate positions $\mathbf{p}_k$ are allowed to leave the image domain; $\Pi_k(\cdot)$ is used *only* to fetch subsequent flow values. We enforce validity *solely at the terminal point*: a trajectory from $s$ to $t$ is considered valid iff the continuous endpoint lies in $\Omega_t$.

**Bidirectional map and landing index.** Combining equation 13–equation 14, define the bidirectional transport

$$\Lambda_{s \to t}(\mathbf{u}) = \begin{cases} \Phi_{s \to t}(\mathbf{u}), & t \geq s, \\ \tilde{\Phi}_{s \to t}(\mathbf{u}), & t < s \end{cases} \quad (15)$$

Given $\mathbf{u} \in \Omega_s$, its terminal continuous location is $\mathbf{q} = \Lambda_{s \to t}(\mathbf{u})$, and its discretized landing pixel is

$$\mathbf{v}^\star = \Pi_t(\mathbf{q}) \in \Omega_t, \qquad \chi_{s \to t}(\mathbf{u}) = \mathbb{1}\big[\mathbf{q} \in \Omega_t\big] \in \{0, 1\} \quad (16)$$

**Pairwise mask and tensor stacking.** The pairwise correspondence mask $M_{s \to t} : \Omega_s \times \Omega_t \to \{0, 1\}$ places a single one at the discretized landing pixel when the terminal point is in-bounds:

$$M_{s \to t}(\mathbf{u}, \mathbf{v}) = \chi_{s \to t}(\mathbf{u}) \, \mathbb{1}\big[\mathbf{v} = \mathbf{v}^\star\big], \qquad \mathbf{u} \in \Omega_s, \ \mathbf{v} \in \Omega_t \quad (17)$$

Consequently, $\sum_{\mathbf{v} \in \Omega_t} M_{s \to t}(\mathbf{u}, \mathbf{v}) \in \{0, 1\}$ for every $\mathbf{u}$, and the self-map reduces to the pixelwise identity: $M_{s \to s}(\mathbf{u}, \mathbf{v}) = \mathbb{1}[\mathbf{v} = \mathbf{u}]$. Stacking $M_{s \to t}$ over all $(s, t)$ yields the rank-6 tensor

$$M \in \{0, 1\}^{T \times h \times w \times T \times h \times w}, \qquad M[s, y, x, t, y', x'] = M_{s \to t}\big((y, x), (y', x')\big) \quad (18)$$

**Remarks.** In practice we adopt nearest-neighbor sampling via $\Pi_t(\cdot)$ in equation 13–equation 14; bilinear interpolation can be substituted without changing the definition in equation 17 because the landing index is discretized by $\Pi_t(\cdot)$. Our construction matches the implementation choice that *intermediate* out-of-bounds positions are permitted, while *only* the *terminal* in-bounds condition $\chi_{s \to t}$ is enforced. Alternatively, the optical flow module can be replaced with other correspondence estimation methods, such as DINOv3 Siméoni et al. (2025) features, for improved performance.

A.5 Dynamic–Appearance Decomposition for a Causal VAE

The causal VAE in Wan Wan et al. (2025) encodes a video using a temporal subsampling factor $r$, producing a *time-compressed* latent sequence. We show that this video latent admits a decomposition into (i) an *appearance* term mainly determined by one frame per $r$-frame block and (ii) a *dynamic* residual. For $r = 4$, the clip's appearance is therefore dominated by $1/4$ of the frames.

**Sampling and per-frame encoding.** Given an $N$-frame video $V = \{V[t]\}_{t=0}^{N-1}$ and an offset $\Delta p \in \{0, \ldots, r\}$, define the sampled frame *sequence*

$$S_r(V, \Delta p) = \big(V[0]\big) \cup \big(V[rj + \Delta p] \,\big|\, j = 0, 1, \ldots, \lfloor (N-1)/r \rfloor - 1\big)$$

Let $\mathscr{E}_{\mathrm{vid}}$ be the video encoder and $\mathscr{E}_{\mathrm{img}}$ the same backbone in image (framewise) mode. The video encoder yields:

$$z(V) = \mathscr{E}_{\mathrm{vid}}(V)$$

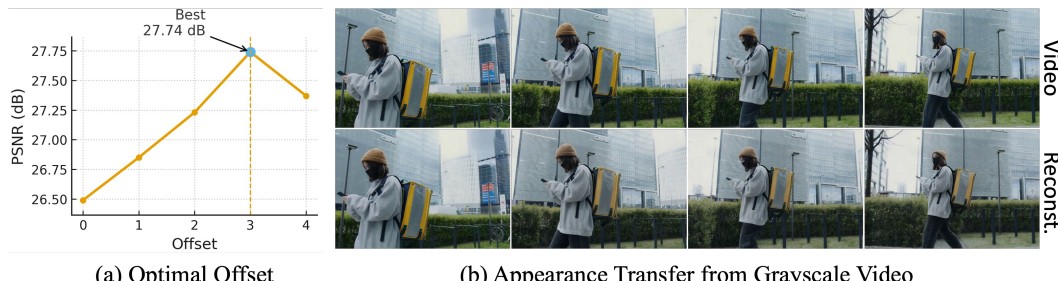

(a) Optimal Offset      (b) Appearance Transfer from Grayscale Video

Figure 9: Optimal Offset and visualization for appearance transfer. (a) The plot for reconstruction PSNR with different offsets; (b) The RGB video and its reconstruction via appearance transfer.

The latents for sampled sequence is obtained by encoding the sampled frames *independently* and stacking them along time:

$$a(V, \Delta p) = \left( \mathscr{E}_{\text{img}}(I) \right)_{I \in S_r(V, \Delta p)}$$

Because the sampled frames are independently encoded, $a(V, \Delta p)$ only contains appearance information. Define the dynamic residual by:

$$z_{\text{dyn}}(V, \Delta p) = z(V) - a(V, \Delta p)$$

**Appearance–dynamics exchange.** For an RGB clip $V_{\text{rgb}}$ and its grayscale counterpart $V_{\text{gray}}$, swapping appearance while preserving dynamics gives:

$$V'_{\text{rgb}}(\Delta p) = \mathscr{D}(z_{\text{dyn}}(V_{\text{gray}}, \Delta p) + a(V_{\text{rgb}}, \Delta p)), \tag{19}$$

$$V'_{\text{gray}}(\Delta p) = \mathscr{D}(z_{\text{dyn}}(V_{\text{rgb}}, \Delta p) + a(V_{\text{gray}}, \Delta p)), \tag{20}$$

where $\mathscr{D}$ denotes the VAE decoder. This implements "subtract source appearance, add target appearance" while keeping the dynamics fixed. Figure 9-(b) demonstrates that the appearance of the RGB video is successfully combined with grayscale dynamics, with only a minor color shift. This observation inspires us to inject the dynamic residual into the value matrices of static reference latents, which addresses the stuttering issue of additional reference inputs in Section 3.2.

**Choosing the offset.** $\Delta p$ selects which frame within each $r$-frame block anchors appearance. We choose it by minimizing cross-reconstruction error:

$$\Delta p^\star = \arg \min_{\Delta p \in \{0, \dots, r\}} \mathcal{L}\left(V'_{\text{rgb}}(\Delta p), V_{\text{rgb}}\right) + \mathcal{L}\left(V'_{\text{gray}}(\Delta p), V_{\text{gray}}\right),$$

where $\mathcal{L}$ can be an $\ell_1$+LPIPS video reconstruction metric. As shown in Figure 9-(a), the reconstruction performance is optimal when $\Delta p = 3$ for $r = 4$. Furthermore, the PCA visualizations of the video latents and the sampled frames exhibit close alignment (Figure 10), indicating that the temporal anchoring is consistent across the two representations. The additional references should be selected from $S_r(V, \Delta p)$, and the optical flow for inconsistency masking and EOG in Section 3.2 should be calculated accordingly for optimal performance.

A.6 DENOISING PROCESS OF WAN I2V MODEL

As in image diffusion models, the early denoising stages in video diffusion primarily establish the coarse global layout and motion of the generated video, while the later stages are dedicated to refining fine-grained details. As illustrated in Figure 10, the primary layout and structure of the entire video are largely determined by around the 20th denoising step. In later steps, the model mainly focuses on the local appearance refinement. The PCA visualizations of the latents align well with our sampled frames with optimal offset.

A.7 NOISE INITIALIZATION

In recent diffusion-based image and video editing works Hertz et al. (2022); Cao et al. (2023); Parmar et al. (2023); Yang et al. (2025b); Li et al. (2024), noise inversion is a prerequisite step for

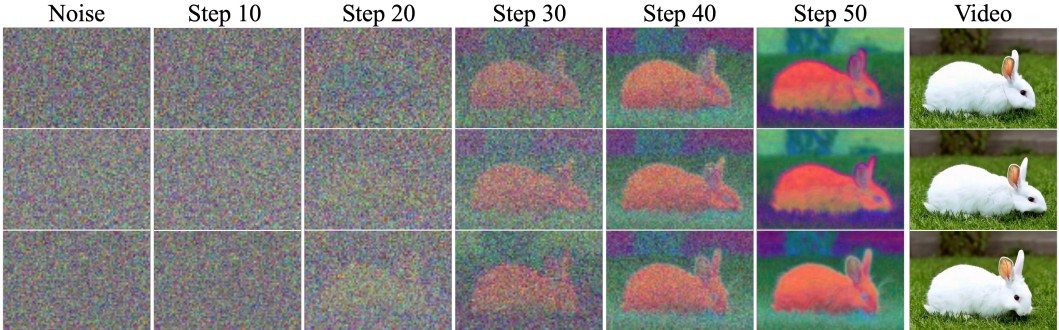

Figure 10: Denoising process of Wan I2V model. We apply PCA to visualize the latents for different frames along the entire denoising process. The corresponding sampled video frames are given.

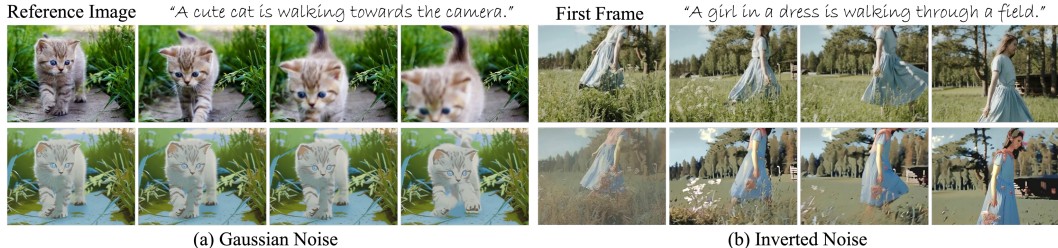

Figure 11: Effect of noise initialization for video stylization with reference. (a) The generated videos share the same initial Gaussian noise. (b) Original video and its stylization via inversion.



Figure 12: **Frequency spectrum of intermediate latents across timesteps.** We apply Fast Fourier Transform (FFT) along the two spatial dimensions. Lower frequencies are concentrated near the center, whereas higher frequencies are distributed toward the corners.

essential content preservation when large-scale training is infeasible. However, we find that noise initialization alone is insufficient to capture the detailed motion and content for video stylization.

Gaussian noise is widely adopted as a standard initialization for generative models Ho et al. (2020); Lipman et al. (2023). Nevertheless, as shown in Figure 11, while the stylized reference image retains the overall semantic layout, it is treated as an **out-of-distribution** example, and the video generated from the same Gaussian noise fails to exhibit realistic dynamic motion. In contrast, the noise obtained through inversion preserves critical dependencies across pixels Staniszewski et al. (2024). Thus, we employ inverted noise for video stylization, which enables partial recovery of camera and object motion; however, content not visible in the first frame remains irrecoverable, and the color is inconsistent with the reference. Therefore, we are seeking additional strategies for more accurate motion and layout preservation in the stylized video.

We further observe that the generated video content is highly sensitive to the reference, and replacing this frame with an edited or stylized one after inversion often introduces significant artifacts. We attribute this issue to the strong attention coupling between the noise and reference latents during the entire denoising stages (Figure 2). Specifically, the noise latent estimated through inversion tends to overfit the distribution of the original reference image. To mitigate this issue, we propose to remove the reference image during inversion, which substantially reduces these artifacts.

## A.8 Frequency Analysis of Intermediate Latents

In the frequency spectrum of the initially inverted noise, all frequency components exhibit similar intensity (Figure 12), characteristic of Gaussian white noise. As the denoising process progresses,

the intensity of high-frequency (HF) components diminishes, while low-frequency (LF) components become more pronounced. This transition occurs most noticeably during the early stages. Therefore, our Indirect High-frequency Compensation (IHC) strategy is only applied in the early timesteps.

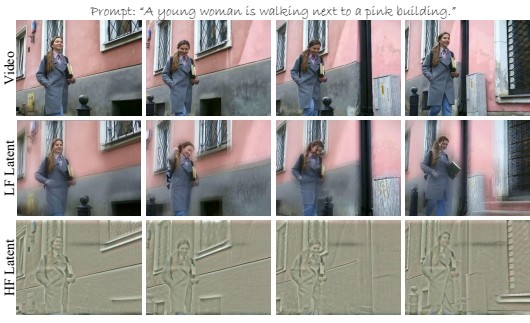

Figure 13: Effects on video reconstruction of LF and HF compensations in PnP inversion.

We observe that different frequency components of the intermediate latents serve distinct roles when applying PnP inversion during inference to enhance content preservation. As Figure 13 illustrates, the low-frequency (LF) components primarily govern the appearance attributes, such as the color distribution, of the generated video. On the contrary, the high-frequency (HF) components predominantly encode layout and motion information. Consistent with characteristics observed in natural images, HF components capture edge and boundary details, which are critical for determining the structural layout. Since the HF component contains less color information, which will not affect the stylized results, we leverage it to constrain the layouts of the stylized videos in our method.

## A.9 LIMITATIONS & FUTURE WORKS

**Time Complexity.** Similar to other training-free image & video editing approaches, FreeViS requires an inversion process to obtain the denoising trajectory, which usually doubles the inference time. In the meanwhile, since FlashAttention Dao (2024) does not provide optimized support for masked attention, the use of masked attention in Section 3 increases the overall time complexity. In practice, FreeViS adds approximately 30% overhead to the stylization time of the AnyV2V* implementation. A straightforward workaround is to use FreeViS for generating original–stylized video pairs prior to the supervised fine-tuning (SFT) of a video diffusion model, thereby compensating for the scarcity of such paired data. Combining with recent advances of auto-regressive video diffusion models, real-time video stylization can be possible.

**Stylization Upper Bound.** The stylization quality of the references is fundamentally constrained by the underlying image style transfer models. FreeViS enforces the pretrained base model to propagate style features from the reference frames across the entire video. However, the base model itself lacks the capability to explicitly parse or interpret style features. As a result, the overall quality of video stylization, both in terms of style similarity and content preservation, is fully determined by the chosen image model. As illustrated in Figures 14 and 15, different image-based models exhibit distinct stylization biases. Future work could explore a mixture-of-experts strategy to selectively combine their complementary strengths according to the style image or video content.

**Optical Flow Estimation.** FreeViS leverages optical flow to identify overlapping regions between reference frames in order to resolve temporal conflicts. However, conventional optical flow estimation methods, such as RAFT Teed & Deng (2020) used in this work, struggle with severe occlusion, leading to temporal inconsistencies in the affected regions. More advanced flow estimation and occlusion-handling techniques could be employed to mitigate this issue. Alternatively, dense and precise feature correspondences extracted with DINOv3 Siméoni et al. (2025) offer a promising direction for constructing more reliable inconsistency masks.

## A.10 FREEVIS WITH VARIOUS IMAGE STYLIZATION METHODS

Leveraging the strong prior of the diffusion model to accomplish high-quality image stylization has become mainstream in recent works Chung et al. (2024); Xu et al. (2025b); Wang et al. (2024a). FreeViS utilizes the high stylization quality of these methods to accomplish video stylization. In this section, we show that different image stylization methods have different strengths, and FreeViS can be generalized to various methods and maintain their strengths. Three state-of-the-art image style transfer models are selected for experiments: StyleID Chung et al. (2024), StyleSSP Xu et al. (2025b), and InstantStyle-plus Wang et al. (2024b).

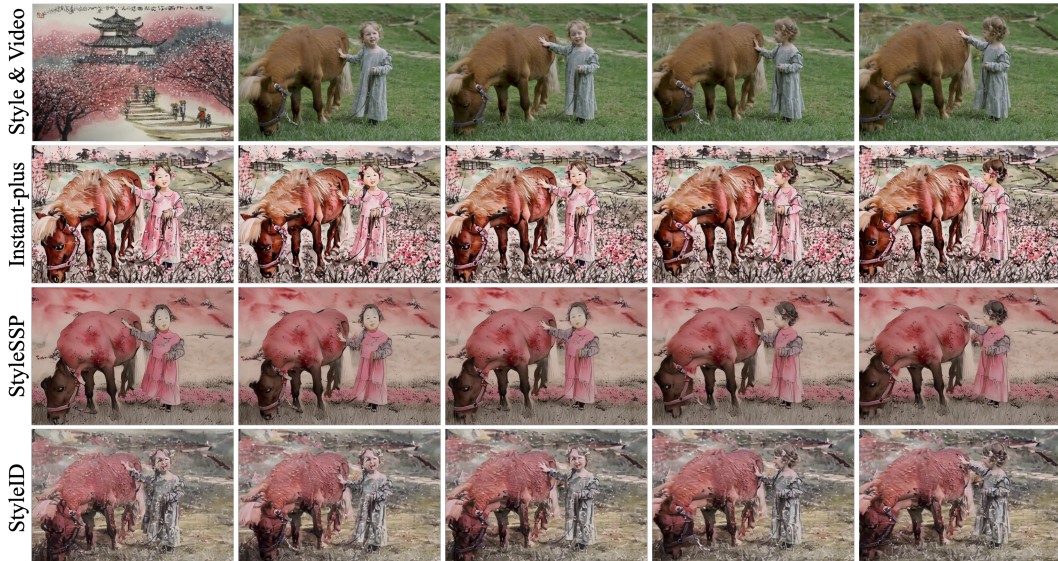

Figure 14: Video stylization results for FreeViS combined with different image style transfer models. "Instant-plus" denotes InstantStyle-plus Wang et al. (2024b).

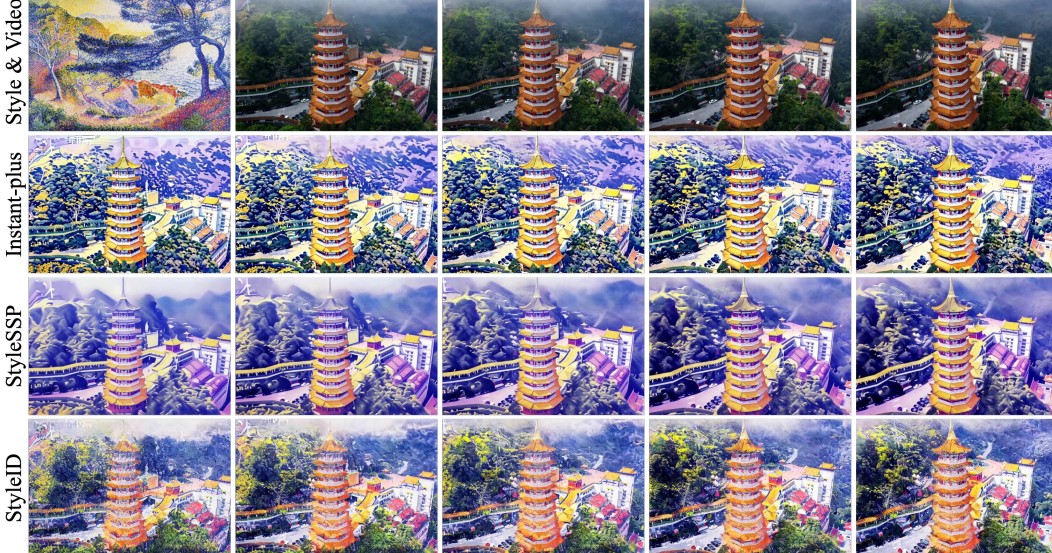

Figure 15: Video stylization results for FreeViS combined with different image style transfer models. "Instant-plus" denotes InstantStyle-plus Wang et al. (2024b).

As shown in Figures 14 and 15, InstantStyle-plus and StyleSSP achieve stronger high-level style alignment, whereas StyleID emphasizes low-level texture preservation. Among them, StyleSSP produces smoother object surfaces and better layout reconstruction compared to the other two methods. InstantStyle-plus provides a more favorable trade-off between visual plausibility and style preservation. Since FreeViS is compatible with all these methods, the underlying image style transfer model can be flexibly substituted to meet the requirements of different scenarios.

## A.11 QUANTITATIVE ABLATION STUDY

The quantitative results for the ablation analysis of each key component in FreeViS are presented in Table 3. Multi-reference integration is demonstrated to be the most effective module, with significant improvements in style preservation and temporal consistency. In Equation 5, $\mathbf{Out}_1^s$ merges the style information from multiple stylized references and contributes to the high-quality style propagation. Besides, $\mathbf{Out}_2^s$ (Equation 6) leverages the spatial-temporal correspondence between patches from

| Method | Stylization Quality | | | | Video Consistency | | | HP ↑ |
|---|---|---|---|---|---|---|---|---|
| | CSD Score ↑ | ArtFID ↓ | FID ↓ | LPIPS ↓ | SC ↑ | MS ↑ | FC ↓ | |
| FreeViS | 0.448 | 33.46 | 21.62 | 0.479 | 0.898 | 0.978 | 0.641 | 4.113 |
| w/o IHC | 0.449 | 33.49 | 21.60 | 0.482 | 0.899 | 0.971 | 0.879 | 4.021 |
| w/o Ext. Refs | 0.314 | 35.27 | 25.78 | 0.317 | 0.755 | 0.975 | 1.374 | 3.523 |
| w/o EOG | 0.409 | 33.89 | 22.16 | 0.463 | 0.886 | 0.974 | 0.753 | 3.973 |

Table 3: Quantitative ablation study for FreeViS. "SC", "MS", "FC", and "HP" denote Style Consistency, Motion Smoothness, Flow Consistency, and Human Preference, respectively.

| Complexity | Training-Free | | | | Training-based | |
|---|---|---|---|---|---|---|
| | FreeViS | AnyV2V* | AnyV2V | TokenFlow | I2VEdit | VACE |
| Time (min) | 4.47 | 3.41 | 11.75 | 1.52 | – | 2.75 |
| Space (GB) | 13.56 | 8.38 | 45.11 | 29.24 | 44.53 | 22.14 |

Table 4: Complexity Analysis of FreeViS and other baselines. Existing training-free stylization methods require additional inversion to obtain the initial noise. **The inversion time has been counted in this table (1.44 mins for FreeViS).** The lengths of evaluation videos are **81 frames** (5 seconds). The runtime for I2VEdit is omitted since it requires inference-time training (more than 1 hour). FreeViS and AnyV2V* share the same base video diffusion model. Note that our AnyV2V* implementation replaces the attention transfer in its original setting by $\{q, k\}$ transfer, since the attention maps are too large, causing OOM issues.

the reconstruction branch to define the layout information, thereby improving the video consistency. Note that the patch correspondences from the reconstruction branch are close to the original video, since the noise latent of the reconstruction branch is directly compensated by PnP inversion.

The functions of Indirect High-frequency Compensation (IHC) and $\text{Out}_2^s$ are relatively similar. Both of them obtain spatial information from the reconstruction branch (or the target latent) to constrain the content structure in the stylized videos. IHC further improves the robustness of content reconstruction for FreeViS in some complex cases that involve significant content changes across video frames, where the inverted noise and $\text{Out}_2^s$ struggle to model the layout, evidenced by the more precise content structure reconstruction and better flow consistency.

Explicit Optical-flow Guidance (EOG) leverages the optical flow extracted from the original video to compensate for the patch correspondence errors in low-saliency regions across temporally distant frames. Since EOG alleviates the local style texture diminishing issue, the overall style patterns of the stylized videos can follow the references to a great extent, with noticeable improvements in style consistency and similarity (CSD Score). More results on the complex texture preservation performance of FreeViS are presented in Figure 17.

## A.12 COMPARISON OF MODEL COMPLEXITY

Training-free video editing methods, which typically require an additional inversion process to initiate noise, often deliver slower inference speeds than training-based methods. VACE and TokenFlow (based on image diffusion models) present fast inference speed but struggle to stylize the video (Figure 4). Due to the manipulation of extra references, FreeViS requires more runtime and memory usage than AnyV2V*, as shown in Table 4. With a slight sacrifice in computational complexity (a 30% increase in inference time), FreeViS achieves significant improvements in stylization quality.

## A.13 HYPER-PARAMETER TUNING

**Multi-Reference Selection.** FreeViS leverages multiple stylized references to address the style propagation issue of existing methods. As we discussed in Section A.7, the diffusion model alone cannot effectively transfer the style patterns from the first reference to the subsequent frames with new content emerging, especially when the camera movements are significant. Given this observation, we introduce the multi-reference integration strategy. Since existing video diffusion models can only process short videos, e.g., 5 seconds (81 frames), the combination of the first and the last frames is able to cover most of the video content. However, this naive implementation shows weak robustness in complex videos, e.g., camera moving forward and backward. Therefore, we

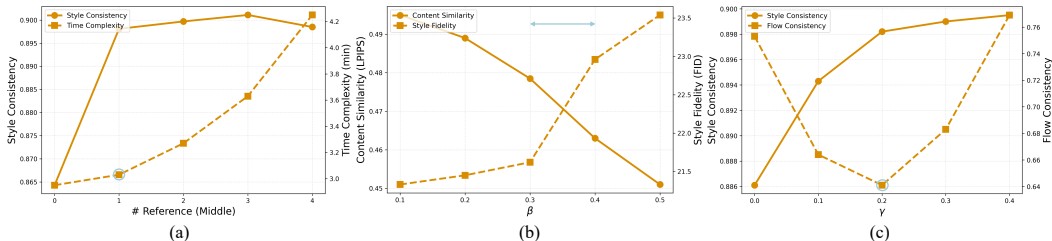

Figure 16: Hyperparameter tuning for FreeViS: (a) Improvements of style consistency ↑ over the number of middle references (first and last references are included as default). The inversion time is not counted in the time complexity; (b) Trade-off of content preservation (LPIPS ↓ with content frames) and style similarity (FID ↓ with style images) with the tuning of $\beta$; (c) The effect of $\gamma$ on Style Consistency ↑ and Flow Consistency ↓.

also include the middle frames as references. Specifically, we uniformly sample the middle frames along the video for more references, where the sampling density is defined by the number of middle references. Note that the sampled middle indices should follow the offset discussed in Section A.5.

As shown in Figure 16-(a), including the last reference (0 middle reference) significantly improves style consistency from 0.755 (see Table 3) to 0.864, and adding a single middle reference further elevates this score to 0.898 with a slight increase in runtime. Note that more middle references improve the robustness of the stylization, but increase the possibility of destroying large style patterns, leading to a slight decrease in style preservation. Empirically, we find that when this middle reference position is at $f//2$ for a video with length $f$, the model achieves the highest stylization quality. The current version of FreeViS adopts a uniform reference sampling strategy, which can be improved to an adaptive strategy based on changes in video content across frames. We leave this possible improvement for future studies.

$\beta$ **Tuning.** In equation 8, $\beta$ controls the strength of reconstruction-guided generation. This strategy can be seen as another form of IHC, as the reconstruction attention maps define the spatial-temporal correlation between pixels in later denoising stages (see figure 2) and can be utilized to indirectly force the stylized video to follow the layout or motion of the original video. However, if this correlation is too strong ($\beta$ is too large), the stylization branch will produce the same semantics as the reconstruction branch in the early denoising stages, leading to the disappearance of stylization features. Also, the newly introduced style features do not strictly adhere to the temporal or geometric layouts of the original videos. Strong reconstruction forcing will cause stylization failures, e.g., missing style textures. Figure 16-(b) shows the trade-off between content similarity and style fidelity with the change of $\beta$. Based on the visual examination, we observe that the highest stylization quality (best trade-off between LIPIS and FID) is accomplished when $\beta \in [0.3, 0.4]$.

$\gamma$ **Tuning.** The EOG module constrains the cross-frame attention areas to compensate for patch correspondence errors in low-saliency regions across temporally distant frames. As in the early denoising stages (first 60% timesteps), the diffusion model requires global information for low-frequency semantic generation, we only apply the EOG strategy in the final stages (last 40% timesteps), where the model primarily focuses on local texture refinement (Figure 10). $\gamma$ defines the strength of this optical-flow guidance in FreeViS. We observe consistent improvements in style preservation as $\gamma$ increases. However, when $\gamma$ exceeds 0.4, there are noticeable localized, blocky jitters in the stylization videos, leading to worse flow consistency. We attribute this issue to the disconnection from surrounding patches in the attention operation inside EOG. Empirically, $\gamma = 0.2$ delivers the best temporal consistency with relatively high-quality style preservation.

## A.14 MORE RESULTS WITH COMPLEX STYLES

FreeViS has the capability of texture preservation, which enables high-quality video stylization with complex styles (emphasizing the strokes). Figure 17 demonstrates the qualitative performance of FreeViS on two classic style images commonly used in recent image style transfer works. The experimental results show that FreeViS successfully preserves local textures during camera and object movements. Please check more results in the supplementary video.

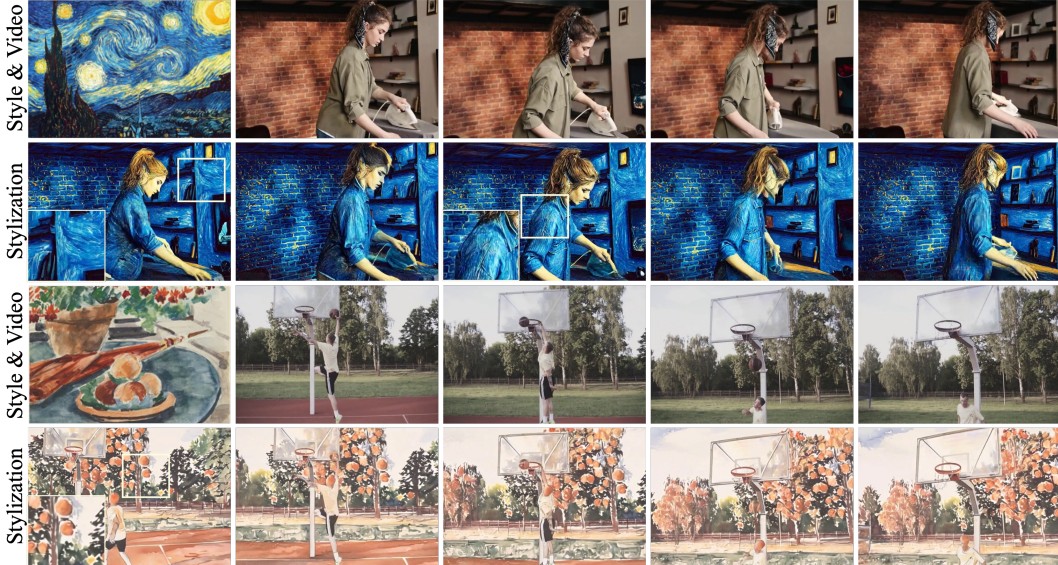

Figure 17: Video stylization results for FreeViS with two classic style images. InstantStyle-plus provides the stylized references for each video. Both videos involve camera and object movements. Local textures are cropped out for better examination (zoom in to check the texture preservation).

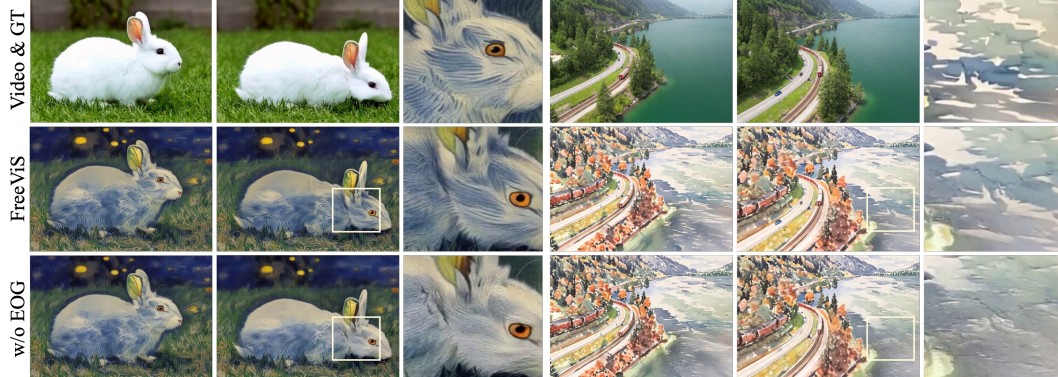

Figure 18: Effect of EOG for texture preservation along object and camera movements. The stylization ground truth (GT) is cropped from the reference image.

| Cutoff Freq. | 0.4 | 0.3 | 0.2 | 0.15 | 0.1 | 0.01 |
|---|---|---|---|---|---|---|
| CSD Score ↑ | 0.4493 | 0.4489 | 0.4481 | 0.3791 | 0.264 | 0.1134 |
| Flow Consistency ↓ | 0.762 | 0.711 | 0.641 | 0.539 | 0.415 | 0.242 |

Table 5: Ablation study for the selection of the cutoff frequency in IHC. This cutoff frequency is defined as a circle around the origin (center), so noise components outside this frequency are passed through the high-pass filter. In this table, from left to right, the stylization latent receives increasing compensation from the target latent (original video).

## A.15 MORE RESULTS WITH EOG

To further demonstrate the effectiveness of EOG for texture preservation, we provide additional qualitative results in Figure 18. The texture-diminishing issue in low-saliency areas along the object or camera movement is alleviated by EOG, as evidenced by sharper, finer-grained textures.

## A.16 CUTOFF FREQUENCY IN IHC

The cutoff frequency of the high-pass filter (HPF) in the proposed IHC module is critical. We observe that if this frequency is too small, the stylized video will be directly dragged back towards the original video, indicated by low style scores and good flow consistency. On the contrary, if the

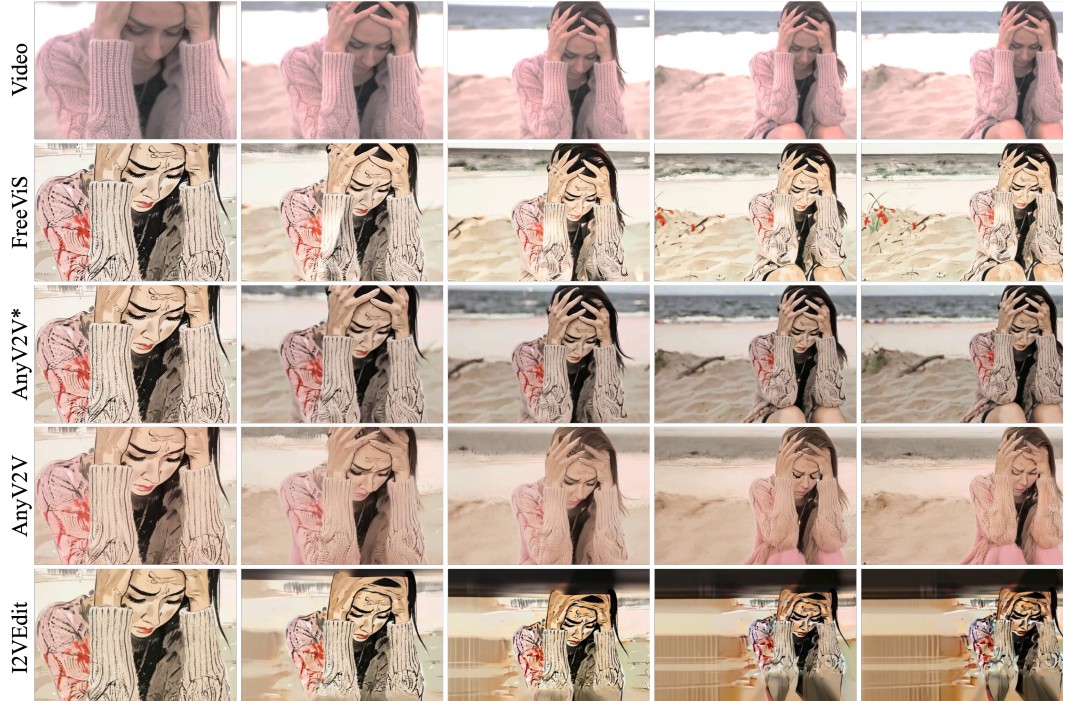

Figure 19: Qualitative comparison of FreeViS with other reference-based video editing methods. All the methods are provided with the same first reference.

frequency is too high, no information is passed through the HPC. Empirically, we set this frequency to be 0.2 for the best trade-off.

### A.17 MORE RESULTS ON VIDEO STYLIZATION

In the main paper, we omit the results of AnyV2V Ku et al. (2024) and I2VEdit Ouyang et al. (2024). For completeness, their results are provided in Figures 19 and 20. Both methods fail to faithfully reconstruct the essential layout of the original video and exhibit the previously discussed propagation errors, primarily due to their reliance on a single reference input. AnyV2V* (our re-implementation of AnyV2V on the Wan model) demonstrates improved layout preservation. In contrast, FreeViS effectively addresses the propagation error and achieves the highest stylization quality.

### A.18 MORE RESULTS ON STYLIZED T2V GENERATION

Additional results on stylized video generation are shown in Figure 21. Consistent with previous observations, StyleCrafter Liu et al. (2024a) demonstrates strong style preservation but struggles with prompt fidelity and motion diversity. In contrast, StyleMaster Ye et al. (2025) achieves better text–video alignment but fails to effectively transfer the target style. The combination of FreeViS with the Wan model provides the best balance between style preservation and prompt fidelity, yielding results that are also more visually plausible.

### A.19 FREEVIS WITH VARIOUS VIDEO DIFFUSION MODELS

To further validate the compatibility of FreeViS, we implement it on another pretrained I2V model, HunyuanVideo-I2V Kong et al. (2024). Owing to its token replacement mechanism, the first reference latent is tightly coupled with the latents of subsequent frames, which leads to severe overfitting during inversion. As a result, standard inversion fails to reconstruct the video and introduces substantial artifacts. In contrast, PnP inversion with compensation proves both more effective and essential, ensuring faithful video reconstruction. We present a qualitative comparison of FreeViS implemented on Wan Wan et al. (2025) and HunyuanVideo in Figure 22. In most cases, the two models achieve comparable stylization performance, while the Wan model demonstrates superior robustness and

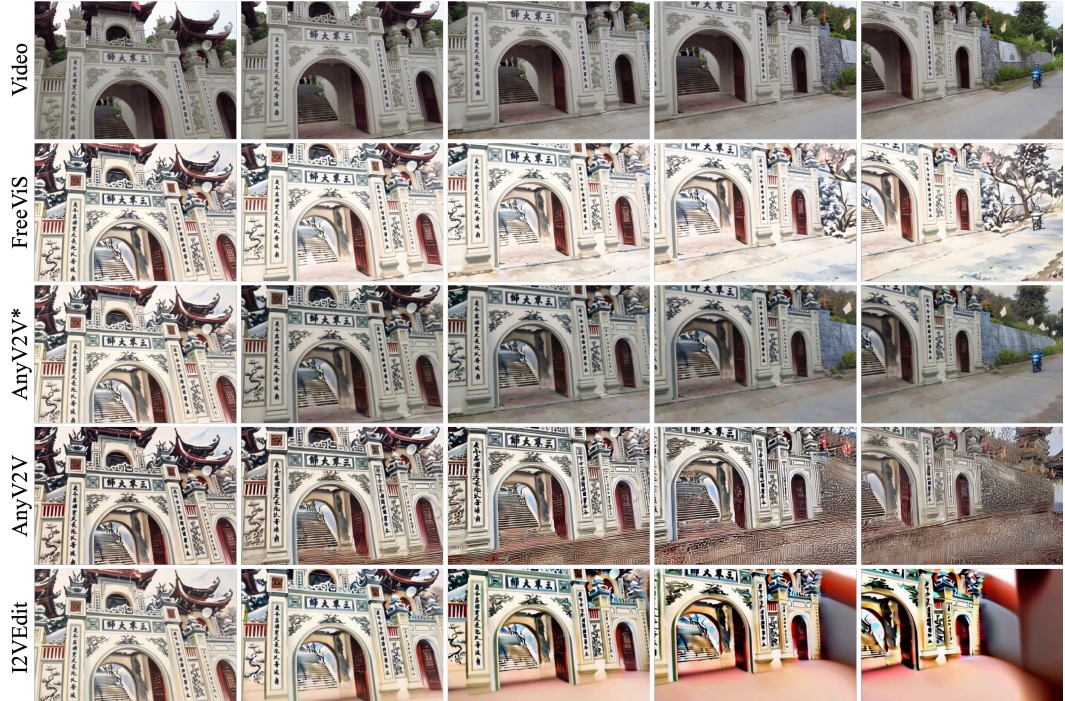

Figure 20: Qualitative comparison of FreeViS with other reference-based video editing methods. All the methods are provided with the same first reference.

temporal consistency. These results highlight the importance of leveraging the strong prior of an advanced video diffusion model to achieve higher stylization quality with FreeViS.

## A.20 USE OF LLMS

The authors first wrote the content of this paper and then gave it to LLM for polishing. The polished content was selectively used to replace the original content, primarily in grammatical aspects.

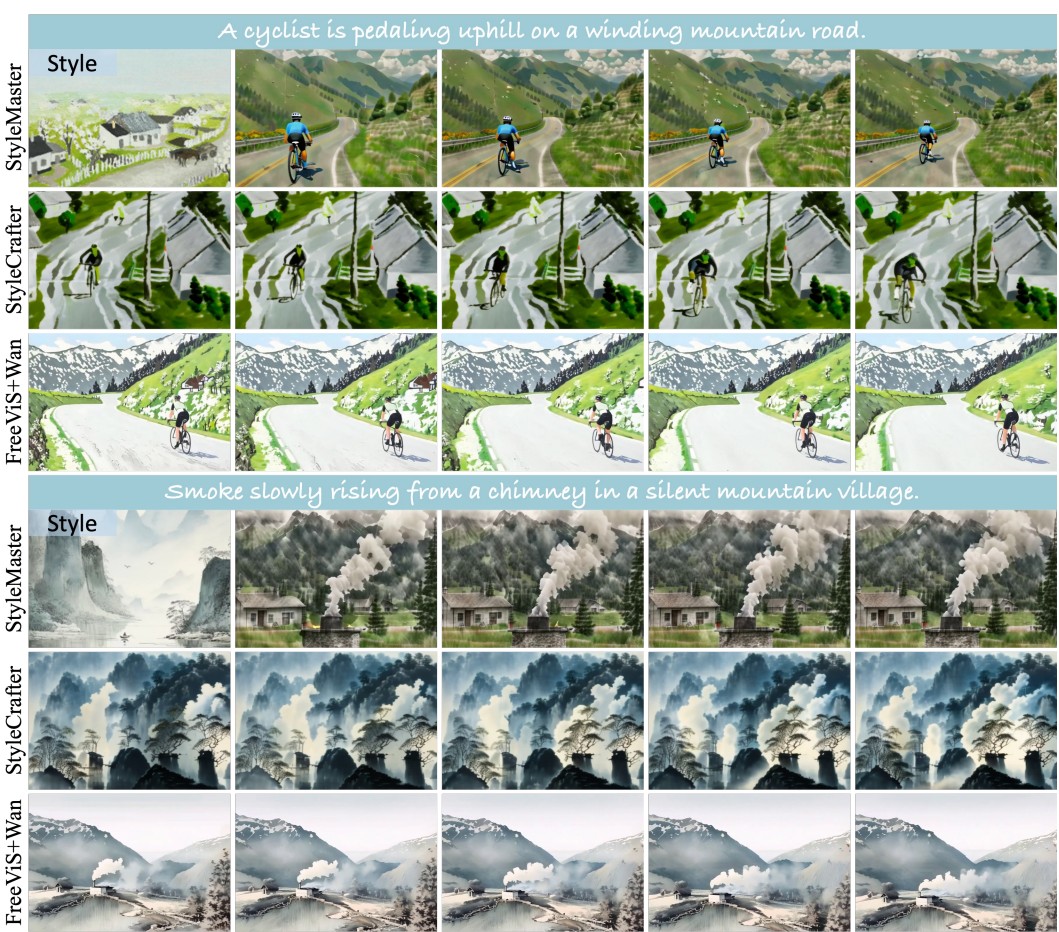

Figure 21: Qualitative comparison of FreeViS with other stylized video generation frameworks.

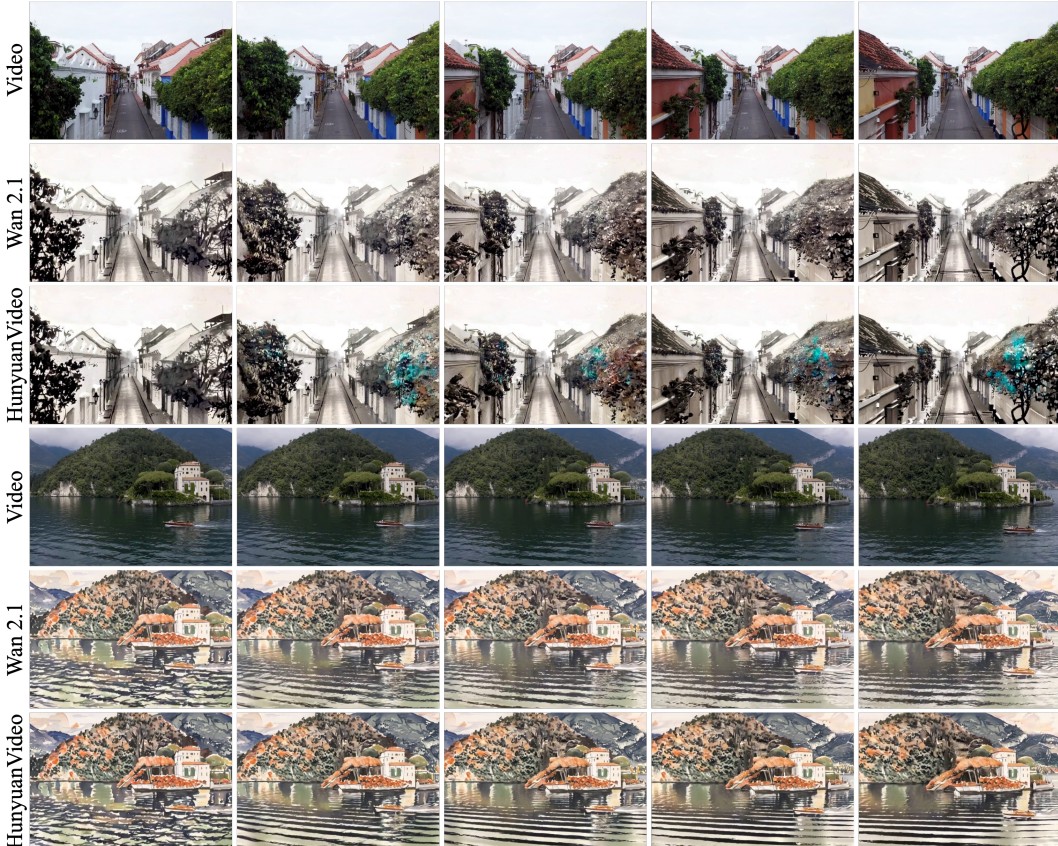

Figure 22: Qualitative comparison of FreeViS combined with different base models: Wan Wan et al. (2025) and HuanyuanVideo Kong et al. (2024). Their stylization results in most cases are similar, while Wan presents smoother reconstruction and better temporal consistency. The third row shows a failure case of HuanyuanVideo in consistency preservation.

