# OpenReview forum: "FreeViS: Training-free Video Stylization with Inconsistent References"
_ICLR.cc/2026/Conference — ICLR 2026 Poster_

### Official Review · Reviewer_nCHs · 2025-10-21

**Soundness:** 3
**Presentation:** 3
**Contribution:** 3
**Rating:** 6
**Confidence:** 3

**Summary:**

This paper proposes FreeViS, a training-free video stylization framework that effectively addresses the propagation issues in reference-based stylization. The key idea is a dual-branch design that incorporates high-frequency compensation, additional inconsistent references, and optical-flow-guided masked attention. The method can be applied to both image-based V2V stylization and stylized T2V generation, demonstrating higher stylization fidelity and competitive temporal consistency compared to prior training-free approaches.

**Strengths:**

- The paper identifies and analyzes the root causes of temporal artifacts in existing training-free stylization, and proposes a well-motivated dual-branch design to address them.
- As a training-free framework, the method is practical.
- The method naturally extends to both video-to-video stylization and stylized text-to-video generation, which increases practical value.
- Compared with existing baselines, the stylized outputs show richer texture, stronger style faithfulness, and fewer propagation errors, supported by convincing qualitative comparisons.

**Weaknesses:**

- While qualitative results are strong, the quantitative metrics in Table 1 and Table 2 are not consistently better than competing methods.
- The ablation study is limited, showing only a single metric and lacking a broader evaluation across other metrics.
- The dual-branch pipeline leads to higher memory usage and longer generation time, yet computation efficiency is only briefly mentioned in the appendix. A quantitative comparison of VRAM usage and runtime is needed.
- Several design choices, such as reference sampling frequency, number of references, and optical-flow strength, are not thoroughly analyzed. A sensitivity study would make the method's robustness and design justification clearer.

**Questions:**

- Why is the video consistency score in Table 1 lower than baselines? Why is style consistency in Table 2 weaker than StyleCrafter? A brief explanation in the paper would improve the reader’s understanding.
- How much slower or heavier is the dual-branch pipeline compared to comparison method? Please provide generation time and VRAM usage comparisons.
- How does performance change when sampling references more or less frequently? Does the choice of image stylization method for reference frames affect results? A sensitivity analysis would strengthen the claims.
- How do other metrics change across ablations, beyond the single metric currently reported? Showing full-metric ablation tables will better support the effectiveness of each component.

---

> ### Author Response · Authors · 2025-11-25
> **Official Comment by Authors [1/2]**
>
> We appreciate your efforts in reviewing this paper and providing us with constructive suggestions. We have added the additional experimental results in the revised submission. Below, we address your questions and concerns.
>
> ---
> ### wrt video consistency score in Tables 1 & 2 (weakness 1 & question 1)
>
> **wrt video consistency score in Table 1.** This is because, for competing methods, such as VACE [Jiang et al. 2025] and TokenFlow [Geyer et al. 2024], failed to perform meaningful stylization (*i.e.* can only modify colors (see Figure 4 and the supplemented video). Their consistency scores do not reflect stylization quality. For example, SC computes the CLIP similarity between frames. Since VACE and Tokenflow fail to produce various style textures in frames based on different semantics, this score will be slightly higher.
>
> In addition, following previous works, such as StyleMaster [Ye et al. 2025], we also evaluate video consistency using MS and FC scores. However, the MS and FC are computed using flow-estimation models pre-trained on natural videos. Some stylized videos might be out of their training distribution, leading to inaccurate flow estimates and scores. We have added this discussion to the revised manuscript.
>
> **wrt video consistency score compared to StyleCrafter in Table 2.** This is because the camera and object motions of the generated videos from StyleCrafter [Liu et al. 2024] are small (see Figure 6 and the comparison results in the supplemented video) and thus is naturally preferred consistency scores. As shown in Figure 6, their results are usually produced from a fixed camera, without background movement, leading to a higher consistency score (CLIP similarity between frames). We have added this discussion to the revised manuscript.
>
> ---
> ### wrt more ablation metrics (weakness 2 & question 4)
>
> We report more ablation results using all metrics in the Table below.
>
> | Method        | CSD Score ↑| ArtFID↓| FID ↓ |LPIPS ↓ | SC  ↑ | MS  ↑  | FC ↓ |
> |:-------------:|:----------:|:------:|:-----:|:------:|:-----:|:------:|:----:|
> | FreeViS       | 0.448      | 33.46  | 21.62 | 0.479  |0.898  | 0.978  |0.641 |
> | w/o IHC       | 0.449      | 33.49  | 21.60 | 0.482  | 0.899 | 0.971  |**0.879** |
> | w/o Ext. Refs | **0.314**      | 35.27  | 25.78 | 0.317  | 0.755 | 0.975  |1.374 |
> | w/o EOG       | **0.409**      | 33.89  | 22.16 | 0.463  | 0.886 | 0.974  |0.753 |
>
> As shown, removing Multi-reference (w/o Ext. Refs) significantly hurts style preservation and temporal consistency. Removing IHC and EOG further harms the robustness of FreeViS on content reconstruction (FC score) and style propagation (SC score). We have added these quantitative results and discussion in Appendix A.11.
>
> ---
> ### wrt computational complexity analysis (weakness 3 & question 2)
> We have added the requested comparison in Appendix A.12 (Table 4) and also report the results in the table below.
>
> | Complexity | FreeViS | AnyV2V* | AnyV2V |TokenFlow| I2VEdit | VACE    |
> |:----------:|:-------:|:-------:|:------:|:-------:|:-------:|:-------:|
> | Time (min) | 4.47    | 3.41    | 11.75  | 1.52    | -       | 2.75    |
> | Space (GB) | 13.56   | 8.38    | 45.11  | 29.24   | 44.53   | 22.14   |
>
> The evaluation is conducted over a set of 5-second videos. Runtime for I2VEdit is omitted as it requires inference-time training (>1 hour).
> Compared to Tokenflow and VACE, although their method are faster, they produce stylized results with poor quality (Fig. 4). While our method does slightly increase in computational complexity (i.e., 30\% over AnyV2V*) due to the dual branch structure, it significantly improves stylization quality (Fig. 1&4 and supplemented video).

---

> ### Author Response · Authors · 2025-11-25
> **Official Comment by Authors [2/2]**
>
> ### wrt reference sampling & stylization and optical-flow strength (weakness 4 & question 3)
>
>
> **wrt performance change with different reference sampling frequency.** In the reference sampling setting, the first and last references are included as default (since in most of the videos, these two frames convey larger content changes). Then we uniformly sample the middle frames of the video for more references, where the number of middle references defines the sampling density.
> | # Reference (middle)| 0       | 1       | 2      | 3       | 4       |
> |:-------------------:|:-------:|:-------:|:------:|:-------:|:-------:|
> | Style consistency ↑  | 0.8643  | 0.8981  | 0.8997 | 0.9011  | 0.8985  |
> | Inference Time  (w/o inversion time)↓    | 2.95    | 3.03    | 3.27   | 3.63    | 4.25    |
>
> Including the last reference (0 middle references) significantly improves style consistency from 0.755 (see Table 3 in the paper) to 0.864, and adding a single middle reference further increases this score to 0.898 with a slight increase in runtime. Note that too many middle references increase the possibility of destroying large style patterns, leading to a slight decrease in style preservation.
>
> **wrt optical flow strengths.** We conduct an ablation study for the optical flow strengths and report results in the table below . The hyperparameter $\gamma$ denotes the strength of the optical-flow guidance (Eqn. 7&8).
> | $\gamma$         | 0.0     | 0.1     | 0.2    | 0.3     | 0.4     |
> |:----------------:|:-------:|:-------:|:------:|:-------:|:-------:|
> | Style Consistency ↑| 0.8861  | 0.8943  | 0.8982 | 0.8990  | 0.8995  |
> | Flow Consistency ↓ | 0.753   | 0.664   | 0.641  | 0.683   | 0.769   |
>
> We choose $\gamma=0.2$ as it is the sweet point between style consistency and flow consistency.
>
> **wrt the choice of image stylization methods.** The image style transfer methods can affect the stylization results of FreeViS, as they provide the frame-wise references. As we discussed in Appendix A.10, FreeViS is generally compatible with various image stylization models, which gives additional flexibility for users to choose.
>
> We have added these results and discussion in Appendix A.13.

---

> ### Comment · Reviewer_nCHs · 2025-11-28
>
> Thank you for your detailed response. The new experiments adequately address the concerns I raised, and I appreciate the effort you put into clarifying the contribution. I will keep my original score.

---

### Official Review · Reviewer_HBzJ · 2025-10-28

**Soundness:** 3
**Presentation:** 3
**Contribution:** 2
**Rating:** 6
**Confidence:** 4

**Summary:**

This paper presents FreeViS, a training-free video stylization framework that leverages a pre-trained image-to-video model, multi-reference integration, and high-frequency compensation to generate stylized videos with rich details and strong temporal coherence.

**Strengths:**

The main innovations are:
1. Training-Free Multi-Reference Integration: A novel pipeline that ingeniously integrates multiple stylized references into a pre-trained I2V model, effectively suppressing error propagation and eliminating flickering without any fine-tuning.

2. High-Frequency Compensation for Layout & Motion Preservation: A dedicated mechanism that utilizes high-frequency signals to constrain content structure and motion dynamics, ensuring superior temporal consistency.

3. Flow-Based Texture Preservation: The innovative use of optical flow motion cues to actively maintain and preserve style textures, enhancing the overall style richness and visual quality.

**Weaknesses:**

1. The proposed method's robustness is intrinsically tied to the accuracy of the optical flow algorithm. Inaccuracies in flow estimation are likely to propagate through the system, potentially leading to residual temporal flickering, which represents a fundamental limitation of the current framework.

2. A significant concern is the ambiguous contribution of the base model versus the novel components. The observed temporal coherence in the stylized videos may be predominantly attributable to the inherent strong temporal consistency of the underlying video generation model, rather than the proposed technique itself. This makes it difficult to isolate and validate the actual improvement brought by the authors' method.

**Questions:**

1. Could you provide more details on how the system handles potential errors from the optical flow computation? Specifically, are there any mechanisms to correct or compensate for flow inaccuracies to prevent visible flickering in the final output?

2. Regarding Equation 2, how was the cutoff frequency for the low-pass filter determined? Could you discuss the sensitivity of the final results to this parameter and whether an ablation study was conducted to guide its selection?

---

> ### Author Response · Authors · 2025-11-25
> **Official Comment by Authors [1/2]**
>
> Thank you for your time in reviewing this paper and providing constructive suggestions. We have added the requested discussions and results in the revised manuscript. Below, we address your concerns and questions.
>
> ---
> ### wrt robustness towards optical flow accuracy (weakness 1 & question 1)
>
> FreeViS uses optical flow only as a soft prior, rather than a hard constraint. Therefore, it is naturally robust to the optical flow accuracy. Specifically, optical flow is used in two places.
>
> **(1)** When integrating multiple references, flow is used merely to identify newly appeared regions in later references (Line 234-243). These regions are large and structurally coherent and do not require precise pixel-level correspondences. Therefore, this module is naturally robust to flow inaccuracies. Small errors in flow do not affect the integration behavior or temporal stability.
> **(2)** When applying flow for texture preservation (*i.e.* Explicit Optical-flow Guidance module), flow is used to restrict the attention computation to relevant areas. This step requires more fine-grained flow, but we achieve high robustness by dilating the flow-guided mask (Line 259-261), allowing attention to operate on a broadened region. This prevents the attention operation from strictly following the exact flow, which can be inaccurate.
>
> However, as discussed in the limitation section, in some extreme cases, such as when the video contains a large amount of occlusions, our method might be affected by the flow algorithms, as such cases remain an open and challenging problem for existing optical flow estimation methods [1]. One potential solution is to use more reference images densely covering the video, but this will increase computational cost.
>
> [1] Luo, Ao, et al. "Flowdiffuser: Advancing optical flow estimation with diffusion models." Proceedings of the IEEE/CVF Conference on Computer Vision and Pattern Recognition. 2024.
>
> ---
> ### wrt effectiveness of proposed techniques over base model (weakness 2)
>
> As discussed in Appendix A.19, we agree that a good base model is important to achieve temporal consistency. Yet a base model alone is far from achieving high-quality stylization. As demonstrated in Figure 7 and Table 3, the proposed techniques are the key factors for the superior performance. Noted that we also re-implemented AnyV2V [Ku et al. 2024] (denoted by AnyV2V\*) using the same base model for fair comparison. The experimental results from Table 1, Figure 4, and the supplemented video suggest the importance of our design. We copied the results from Table 1 in the paper and pasted them below for your reference.
> | Method        | CSD Score ↑| ArtFID↓| FID ↓ |LPIPS ↓ | SC  ↑ | MS  ↑  | FC ↓ |
> |:-------------:|:----------:|:------:|:-----:|:------:|:-----:|:------:|:----:|
> | FreeViS       | 0.448      | 33.46  | 21.62 | 0.479  |0.898  | 0.978  |0.641 |
> |AnyV2V\*       | 0.270      | 34.81  | 27.59 | 0.218  | 0.675 | 0.983  |1.103 |

---

> ### Author Response · Authors · 2025-11-25
> **Official Comment by Authors [2/2]**
>
> ### wrt cutoff frequency for low-pass filter (question 2)
>
> We do select the cutoff frequency based on its performance, where we conducted an experiment to evaluate its effect. We include the experiment results in the Table below. In general, we found that if this frequency is too small, the stylized video will be directly dragged back towards the original video, indicated by low style scores yet good flow consistency. On the contrary, if this frequency is too high, no information passes through the high-pass filter. Based on the results, we set this frequency to be 0.2, which gives the best trade-off.
>
> |  Cutoff Freq.      | 0.4     | 0.3     | 0.2    | 0.15    | 0.1     | 0.01    |
> |:------------------:|:-------:|:-------:|:------:|:-------:|:-------:|:-------:|
> | CSD Score ↑        | 0.4493  | 0.4489  | 0.4481 | 0.3791  | 0.2640  | 0.1134  |
> | Flow Consistency ↓ | 0.762   | 0.711   | 0.641  | 0.539   | 0.415   | 0.242   |
>
> Regarding sensitivity, we can see that within $[0.15, 0.3]$ the variations in terms of stylization quality (CSD Score) and video consistency (Flow Consistency) are small. The performance becomes more sensitive outside this region. We have added this table (Table 5) to the revised manuscript.

---

### Official Review · Reviewer_razj · 2025-10-30

**Soundness:** 3
**Presentation:** 3
**Contribution:** 3
**Rating:** 6
**Confidence:** 3

**Summary:**

This paper proposes FreeViS, a training-free video stylization framework that aims to solve temporal inconsistency (flickering) in frame-by-frame methods and propagation error in single-reference I2V-based methods. The key innovation is integrating multiple stylized reference frames into a pretrained I2V model through three main technical components: 1. Indirect High-Frequency Compensation (IHC) that preserves layout/motion while maintaining stylized colors, 2. an isolated attention mechanism with dynamic injection and optical flow-based masking to handle inconsistent references, and 3. Explicit Optical Flow Guidance (EOG) to preserve textures in low-saliency regions. Experiments demonstrate improvements over existing methods in both stylization quality and temporal consistency.

**Strengths:**

- The IHC mechanism is well-motivated by frequency analysis showing high frequency components encode layout/motion while low frequency components control appearance.
- Solve the multi-reference challenges, dynamic injection addresses stuttering, flow-based masking resolves inconsistencies.
- The qualitative results in Figures 1, 4, 5, 6, 16, and 17 are very convincing and show a clear improvement in style propagation and temporal consistency over baselines.
- The quantitative evaluations and ablation study  are comprehensive and support the authors' claims.
- Clear improvements over baselines in both video-to-video stylization and stylized text-to-video generation tasks

**Weaknesses:**

- Missing hyperparameter ablation studies and analysis, the method combines three attention modes using hyperparameters β and γ (Eq. 8), but no ablation or analysis is provided. How sensitive is the final video quality to these parameters? Additionally, γ is only non-zero in the "final stage" but this stage is not clearly defined, what threshold determines this?
- No inference runtime comparisons with baselines are provided. This makes it difficult to assess the practical computational cost relative to quality gains.
- Reference frame selection not justified or ablated. The paper uses first, middle, and last frames, calling this an "empirically effective choice" without systematic study. Why specifically these 3 frames? How does this compare to different numbers of references? What is the performance vs. efficiency trade-off with more references?

**Questions:**

- Can you provide ablation studies on hyperparameters β and γ? How is the "final stage" for γ activation defined?
- Can you provide inference runtime comparisons with baselines?
- Can you justify the choice of 3 reference frames and provide ablations with different numbers/positions of references?

---

> ### Author Response · Authors · 2025-11-25
> **Official Comment by Authors [1/2]**
>
> Thanks a lot for your efforts in reviewing this paper and providing constructive suggestions. We have added all the requested results in the new submission. Below, we address your concerns and questions.
>
> ---
> ### wrt hyperparameter ablation study (weakness 1 & question 1)
> We conduct ablation studies for hyperparameter $\beta$, $\gamma$ in the tables below.
>
> **wrt ablation study on hyperparameter  $\beta$ .**
> | $\beta$       | 0.1           | 0.2           | 0.3           | 0.4           | 0.5           |
> |:-------------:|:-------------:|:-------------:|:-------------:|:-------------:|:-------------:|
> | LPIPS    ↓     | 0.495         | 0.489         | 0.479         | 0.463         | 0.451         |
> | FID    ↓       | 21.33         | 21.45         | 21.62         | 22.96         | 23.54         |
>
>
> LPIPS measures the content similarity between the stylized and content videos, while FID measures style fidelity. $\beta$ controls the strength of reconstruction-guided generation (Equation 6). Large  $\beta$ can harm stylization, e.g., missing style textures. This table shows there is a trade-off between content similarity and style fidelity. We set $\beta \in [0.3,0.4]$ for the best stylization quality and content fidelity.
>
> **wrt ablation study on hyperparameter  $\gamma$ .**
> | $\gamma$         | 0.0     | 0.1     | 0.2    | 0.3     | 0.4     |
> |:----------------:|:-------:|:-------:|:------:|:-------:|:-------:|
> | Style Consistency ↑| 0.8861  | 0.8943  | 0.8982 | 0.8990  | 0.8995  |
> | Flow Consistency ↓ | 0.753   | 0.664   | 0.641  | 0.683   | 0.769   |
>
> While there is a consistent improvement in style preservation as $\gamma$ increases, large $\gamma$ might also hurt flow consistency. We choose  $\gamma=0.2$  as it is the sweet point between style consistency and flow consistency.
>
> **wrt $\gamma$  Activation.** "Final stage" for γ activation is defined as the last 40\% timesteps of the denoising process. For example, if the total number of denoising steps is 50, then the threshold for the "Final stage" is timestep $>30$.
>
> ---
> ### wrt runtime comparison (weakness 2 & question 2)
> We report the runtime below and compare our method with baselines.
>
> |   | FreeViS | AnyV2V* | AnyV2V |TokenFlow| I2VEdit | VACE    |
> |:----------:|:-------:|:-------:|:------:|:-------:|:-------:|:-------:|
> | Time (min) | 4.47    | 3.41    | 11.75  | 1.52    | -       | 2.75    |
>
> The inversion time is counted in this table (1.44 mins for FreeViS). The evaluation is conducted over a set of 5-second videos. The runtime for I2VEdit is omitted because it requires inference-time training (>1 hour). Compared to Tokenflow and VACE, although their methods are faster, they produce stylized results of poor quality, with missing style textures (Fig. 4). While our method does slightly increase runtime (i.e., 30% over AnyV2V*), it significantly improves stylization quality (as shown in Fig. 4&5 and the supplemented video).

---

> ### Author Response · Authors · 2025-11-25
> **Official Comment by Authors [2/2]**
>
> ### wrt reference frame selection (weakness 3 & question 3)
>
> **wrt number** We provide a detailed study on reference frame selection in the table below. Specifically, the first and last references are always included (as they are natural anchors in a video). We then uniformly sample the middle frames in between for more references and cover diverse positions. Note that including more reference frames will increase inference time.
>
> | # Reference (middle)| 0       | 1       | 2      | 3       | 4       |
> |:-------------------:|:-------:|:-------:|:------:|:-------:|:-------:|
> | Style consistency ↑  | 0.8643  | 0.8981  | 0.8997 | 0.9011  | 0.8985  |
> | Inference Time (min) ↓     | 2.95    | 3.03    | 3.27   | 3.63    | 4.25    |
>
> The inversion time is not counted in the inference time in this table. Including the last reference (0 middle references) significantly improves style consistency from 0.755 (see Table 3) to 0.864, and adding a single middle reference further elevates this score to 0.898 with a slight increase in runtime. Further including more references has only limited or no improvement over the style consistency, but at the price of constantly increasing inference time. Using 3 frames gives the best trade-off between stylization quality and runtime efficiency. This might be because 3 frames are enough to cover most of the content for a 5-second video (the maximum video length that existing video diffusion models can stably process).
>
> **wrt position** Then, we can decide the position for this single middle reference given a video with length of $f$. The experimental results shown below indicate that when the reference position is at $f/2$, the model achieves the highest stylization quality.
> | Pisition (middle frame) | $f/4$      | $f/2$      | $3f/4$      |
> |:-------------------:|:-------:|:-------:|:------:|
> | Style consistency ↑  | 0.8879  | 0.8981  | 0.8943 |
>
> For more discussion about the reference selection, please refer to Appendix A.13 in the revised manuscript.

---

### Official Review · Reviewer_NmoF · 2025-10-31

**Soundness:** 3
**Presentation:** 3
**Contribution:** 2
**Rating:** 4
**Confidence:** 4

**Summary:**

The paper introduces FreeViS, a training-free video stylization framework that addresses the challenges of temporal consistency and style richness in video generation. Instead of relying on paired video data or fine-tuning, FreeViS integrates multiple stylized reference images into a pretrained image-to-video (I2V) model. Its design incorporates indirect high-frequency compensation to refine content layout and motion fidelity, multi-reference integration to reduce flickering and propagation errors, and explicit optical-flow guidance to maintain style textures in low-saliency regions. The authors claim that FreeViS outperforms recent baselines in both stylization quality and coherence.

**Strengths:**

1. Originality: The idea of combining inconsistent stylized references with a pretrained I2V model in a training-free manner is practical and relevant, especially for low-resource settings.
2. Quality: The method is well-motivated and shows promising results in terms of stylization fidelity and temporal coherence. The integration of high-frequency compensation and flow cues is conceptually sound.
3. Clarity: The paper is generally easy to follow, with clear motivation and methodology. Figures and examples help illustrate the core ideas.
4. Significance: FreeViS addresses a real-world need for efficient and high-quality video stylization without the burden of training, making it potentially impactful for content creators and researchers.

**Weaknesses:**

1. Limited Novelty: Several components (e.g., AdaIN, cross-frame attention, flow-based guidance) have been used in prior works like Style Injection in Diffusion, StyleID, and Text-to-Video Zero. The novelty lies more in the integration to video diffusion model than in individual techniques.
2. Unclear Use of Multiple References: Although the paper claims to use multiple stylized references, the experiments and visualizations consistently show only a single style image. There is no explanation of how multiple references are selected, fused, or weighted.
3. Insufficient Evaluation of Key Components: The Indirect High-Frequency Compensation is mentioned but not thoroughly analyzed. The ablation image lacks accompanying discussion, making it hard to assess its contribution.
4. Optical Flow Reliability: The use of flow in low-saliency regions is promising, but flow errors can degrade detail preservation, especially for moving objects. Figure 7 shows only one example, and it’s unclear whether it involves motion.
5. Style Diversity: The method is tested on relatively simple styles. It’s unclear how well it generalizes to complex or abstract styles with texture details.

**Questions:**

1. Could you clarify how multiple stylized references are used in practice?
2. How does the method handle inaccurate optical flow, especially in fast-moving scenes or occlusions?
3. Have you tested FreeViS on more diverse styles with texture details (e.g. oil painting with strokes) or content with texture details (e.g. hair)? If so, how does it perform in terms of fidelity and consistency?

---

> ### Author Response · Authors · 2025-11-25
> **Official Comment by Authors [1/2]**
>
> Thank you for your detailed review and constructive comments! Please note our top-level comment with additional experimental results in the revised manuscript. In the comment below, we address your questions and concerns.
>
> ---
> ### wrt novelty (weakness 1)
>
> **wrt several components used in prior works.** We'd like to clarify that we do not claim AdaIN and cross-frame attention as our novelty. Our major contribution is to build a training-free stylization framework that leads to superior quality compared to prior works. This is enabled by **(1)** a novel mechanism (*i.e.*, training-free multi-reference integration) that integrates multiple stylized references into a pre-trained I2V model, effectively suppressing propagation error and eliminating flickering without any fine-tuning, **(2)** a dedicated design (*i.e.*, high-frequency latent compensation) that utilizes high-frequency signals to constrain content structure and motion dynamics without affecting the stylized information, and **(3)** a new usage of optical flow motion cues to constrain the spatial-temporal attention area, which differs significantly from previous flow-based warping strategy, actively maintaining and preserving style textures, enhancing the overall style richness and visual quality. These core technical contributions have never been explored in prior works.
>
> **wrt integrating components into video diffusion models.** Beyond these core designs, we'd like to mention that building such a training-free stylization system upon existing video models is also non-trivial. As demonstrated in Appendix A.7 & A.8, directly applying reference-based stylization by inversion fails to exhibit realistic dynamic motion. Other works, such as AnyV2V *[Ku et al. 2024]*, suffer from severe propagation error (see Figures 1 & 4, and the supplemented video). In contrast, our work carefully integrates these proposed designs, which significantly improves stylization quality and temporal consistency.
>
> ---
> ### wrt usage of multiple references (weakness 2 & question 1)
>
> **wrt reference selection.** We uniformly select reference images over the time duration. For a video of length \(F\), we select frames with indices  $ \in $ \{1, F//2, F\} as the references, which provide the best trade-off between stylization quality and inference speed (see Appendix A.13 for more analysis).
>
> **wrt reference fusion.** First, these references are independently encoded by the VAE and concatenated with the noise latent along the sequence. Each reference token shares the same positional embedding corresponding to its frame latent counterpart, which guarantees correct spatial and temporal propagation of stylized features. Then, inside the video diffusion model, these reference tokens are processed using Eqns. (3)--(7) for each denoising step. In this way, information from different references is fused by the video diffusion model.
>
> **wrt reference weighting.** There is no weighting operation involved in FreeViS (*i.e.*, we weight them equally). An optical-flow-based masking strategy is implemented to handle the overlapping conflict in Eqn. (5). We have included these discussions in the revised manuscript.
>
> ---
> ### wrt more evaluation of key components (weakness 3)
>
> **wrt more analysis of Indirect High-Frequency Compensation (IHC).** We have included more comprehensive quantitative comparisons in Appendix A.11 (Table 3), which evaluate the effectiveness of Indirect High-Frequency Compensation (IHC) as well as other components. Here we provide the results for IHC:
>
> | Method  | CSD Score ↑ | LPIPS ↓ | SC ↑  | MS ↑  | FC ↓  |
> |:-------:|:-----------:|:-------:|:-----:|:-----:|:-----:|
> | FreeViS | 0.448       | 0.479   | 0.898 | 0.978 | 0.641 |
> | w/o IHC | 0.449       | **0.482** | 0.899 | 0.971 | **0.879** |
>
> Using IHC consistently improves results across metrics related to content similarity. This is because IHC improves the robustness of content reconstruction in complex cases, which involve significant content changes across video frames, where the inverted noise and reconstruction-guided attention (Eqn. 6) struggle to model the layout. This is also validated in the **ablation image** (Figure 7), where the model fails to reconstruct the roof structure in the last frame (which is significantly different from the first frame) after removing IHC, leading to content misalignment with the original video. We have added these discussions to the paper.

---

> ### Author Response · Authors · 2025-11-25
> **Official Comment by Authors [2/2]**
>
> ---
> ### wrt optical flow reliability (weakness 4 & question 2)
> **wrt handle inaccurate optical flow.** Our model does not rely heavily on flow accuracy. It is intentionally designed to minimize reliance on flow accuracy, *i.e.* uses flow as a soft prior, instead of a hard constraint. Because we do not explicitly warp pixels, features, or align noise using flow (which was used in prior works [1, 2]), flow errors are not directly propagated into the model. Specifically, the multi-reference integration module uses flow to mark new-content regions (lines 234 to 243), which are large and structurally coherent, rather than pixel-level correspondences, making the module naturally robust to minor flow inaccuracies. Although the Explicit Optical-flow Guidance (EOG) requires more fine-grained flow, we further ensure robustness by dilating the flow-guided mask (lines 259 to 261), allowing attention to operate on a broadened region. This prevents the attention operation from strictly following the exact flow, which can be inaccurate.
>
> **wrt more examples of moving objects.** We provide two more examples with object and camera motions in Appendix A.15 (Figure 18). The experimental results show that the texture-diminishing issue in low-saliency areas along the object or camera movement is alleviated by the proposed method, as evidenced by sharper, finer-grained textures.
>
> However, as we discussed in the limitation section, in some extreme cases, such as when the video contains a large amount of occlusions, existing optical flow estimation methods may fail to correctly detect all new content (this is still an open problem for flow estimation [3]), leading to inconsistencies in local areas. Densifying the middle reference sampling (more reference images) can help, but it requires more computation. Alternatively, more advanced occlusion-handling techniques could be employed to mitigate this issue. Despite this limitation, FreeViS is generally robust to common cases and shows significant improvement over existing methods, as shown in the supplemented video.
>
> [1] Liang, Jingyun, et al. "Movideo: Motion-aware video generation with diffusion model." European Conference on Computer Vision. Cham: Springer Nature Switzerland, 2024.
>
> [2] Burgert, Ryan, et al. "Go-with-the-flow: Motion-controllable video diffusion models using real-time warped noise." Proceedings of the Computer Vision and Pattern Recognition Conference. 2025.
>
> [3] Luo, Ao, et al. "Flowdiffuser: Advancing optical flow estimation with diffusion models." Proceedings of the IEEE/CVF Conference on Computer Vision and Pattern Recognition. 2024.
>
> ---
> ### wrt style diversity (weakness 5 & question 3)
> **More Experiment on Complex Styles with Texture Details.** We added additional visual results with complex styles in Appendix A.14 (Figure 17). The results show that FreeViS is capable of propagating and preserving detailed style patterns across frames while maintaining video consistency, even when the style becomes complex. Note that the function of FreeViS is to propagate the style patterns in the references to other frames. Therefore, the stylization performance is upper-bounded by the image style transfer model.

---

### Author Response · Authors · 2025-11-26

We sincerely thank all reviewers (**R1**(NmoF), **R2**(razj), **R3**(HBzJ), and **R4**(nCHs)) for providing constructive feedback that helped us improve the paper. We are encouraged that the reviewers think:

* Our approach is novel (**R3**), well-motivated (**R1**, **R2**, **R4**), and practical (**R1**, **R4**);

* The performance improvement is clear and convincing  (**R2**, **R4**);

* Experimental evaluations are comprehensive (**R2**), and key components are effective (**R3**);

* The paper is easy to follow (**R1**).

We have been working diligently on improving the paper on several fronts, addressing the critiques.

Please note that we have included figures, tables, and discussions for the suggested experiments in the revised submission (marked in blue). We address questions and concerns for each reviewer in the comments below.

---

### Meta-Review · Area_Chair_uFTH · 2026-01-06

**Summary:**

The initial reviews were split between 4~6. The primary concerns focused on limited technical novelty, lack of clarity regarding multi-reference handling, and the potential fragility of the optical flow dependency. Reviewers also requested more rigorous evidence regarding hyperparameter sensitivity, computational overhead, and the effectiveness of the Indirect High-Frequency Compensation compared to simply relying on the base Image-to-Video model's inherent stability.

**Reviewer Concerns:**

Addressed:
1. The authors clarified the selection process (uniform sampling) and the fusion mechanism (independent VAE encoding and shared positional embedding). This resolved Reviewer NmoF and razj concerns about the "black box" nature of reference usage.
2. The rebuttal provided comprehensive ablation tables for hyper-parameters, which demonstrated clear trade-offs between style fidelity and content similarity.
3. Contributions: new quantitative comparisons are made to prove the contributions.

Still outstanding:
1. While the authors provided runtime data, most reviewers view this as a justified trade-off for the gain in quality.
2. The authors admitted that extreme occlusions still pose a challenge for flow-based methods.

**Reviewer Scores:**

Reviewer NmoF: The detailed explanation of the multi-reference integration and the additional ablation on IHC should addressed most concerns. So 4 -> 6.

Reviewer razj: This reviewer was already leaning positive; the detailed response to their three questions (hyperparameters, runtime, and reference selection) likely would have pushed them to a strong accept. 6 -> 8

Reviewer HBzJ: No change.

Reviewer nCHs: No change.

---

### Decision · Program_Chairs · 2026-01-26

Accept (Poster)